# The USP12/46 deubiquitinases protect integrins from ESCRT-mediated lysosomal degradation

Kaikai Yu [ID][1], Guan M Wang [ID][1], Shiny Shengzhen Guo [ID][1], Florian Bassermann [ID][2,3,4,5] & Reinhard Fässler [ID][1✉]

## Abstract

**The functions of integrins are tightly regulated via multiple mechanisms including trafficking and degradation. Integrins are repeatedly internalized, routed into the endosomal system and either degraded by the lysosome or recycled back to the plasma membrane. The ubiquitin system dictates whether internalized proteins are degraded or recycled. Here, we use a genetic screen and proximity-dependent biotin identification to identify deubiquitinase(s) that control integrin surface levels. We find that a ternary deubiquitinating complex, comprised of USP12 (or the homologous USP46), WDR48 and WDR20, stabilizes β1 integrin (Itgb1) by preventing ESCRT-mediated lysosomal degradation. Mechanistically, the USP12/46-WDR48-WDR20 complex removes ubiquitin from the cytoplasmic tail of internalized Itgb1 in early endosomes, which in turn prevents ESCRT-mediated sorting and Itgb1 degradation.**

**Keywords** Ubiquitination; Integrin; DUB; USP12/USP46; ESCRT
**Subject Categories** Cell Adhesion, Polarity & Cytoskeleton; Membranes & Trafficking; Post-translational Modifications & Proteolysis

## Introduction

Integrins are α/β heterodimers that mediate cell adhesion between cells and to the extracellular matrix (ECM) proteins (Hynes, 2002). The function of integrins is tightly regulated, on the one hand, by changing the conformational state that turns ligand binding on and off (Calderwood et al, 2013; Moser et al, 2009), and on the other hand, by adjusting the surface location and levels through an endosomal sorting process that dictates whether the integrins are recycled back to the cell surface or delivered to lysosomes for degradation (Moreno-Layseca et al, 2019). Considering that the approximate half-life of Itgb1-class integrins is 24–48 h, their cell surface residence time is 10 min, and the recycling from and back to the plasma membrane is around 20 min (Böttcher et al, 2012;

Dozynkiewicz et al, 2012; Moreno-Layseca et al, 2019), it can be assumed that Itgb1-class integrins undergo numerous cycles of endocytosis and recycling during their lifespan before they are degraded in lysosomes.

The ubiquitin system is a labeling system that marks proteins for different proteolytic fates, such as integrins that are determined for lysosomal degradation. Integrins and other cell surface proteins designated for internalization are ubiquitin-tagged at lysine residues in their cytoplasmic tail (Clague et al, 2012). The removal of the ubiquitin tags by specific deubiquitinases (DUBs) in early endosomes directs proteins into the recycling pathway and back to the cell surface (Clague et al, 2012; Komander et al, 2009). Proteins, which retain the ubiquitin tag are recognized by the endosomal sorting complex required for transport (ESCRT) complex, sequestered into microdomains and internalized as intraluminal vesicles (ILVs) leading to the formation of multivesicular bodies (MVBs), also known as late endosomes (Hanson and Cashikar, 2012). MVBs/late endosomes either mature into lysosomes in which transmembrane membrane proteins on ILVs are degraded by lysosomal proteases, or fuse with the plasma membrane which leads to the extracellular release of their cargo, including the ILVs as exosomes (Huotari and Helenius, 2011; Saftig and Klumperman, 2009).

Previous studies have shown that the binding of α5β1 integrin to soluble fibronectin (FN) induces integrin cytoplasmic tail ubiquitination, internalization, and the degradation of the integrin (Kharitidi et al, 2015; Lobert et al, 2010). It has also been demonstrated that internalized Itgb1 recruits the SNX17-retriever complex, which leads to the retrieval and recycling of integrins (Böttcher et al, 2012; McNally et al, 2017; Steinberg et al, 2012). Indeed, SNX17-binding-deficient Itgb1-tail mutants fail to recycle and are degraded in the lysosome. They can be rescued from degradation upon additionally substituting the α5β1 integrin tails' lysines for non-ubiquitinatable arginines, which led to the hypothesis that SNX17 fulfils two functions: on one hand, it recruits DUB(s) to deubiquitinate the Itgb1 tail (Böttcher et al, 2012), and on the other hand it recruits the retriever complex to retrieve and recycle integrins. USP9X has been identified to bind SNX17 and deubiquitinate centriolar satellite proteins required for ciliogenesis (Wang et al, 2019). Although integrin deubiquitination has not been investigated in this report, Kharitidi and colleagues

[1]Department of Molecular Medicine, Max Planck Institute of Biochemistry, Martinsried, Germany. [2]Department of Medicine III, TUM School of Medicine and Health, Technical University of Munich, Munich, Germany. [3]TranslaTUM, Center for Translational Cancer Research, Technical University of Munich, Munich, Germany. [4]Deutsches Konsortium für Translationale Krebsforschung (DKTK), Heidelberg, Germany. [5]Bavarian Cancer Research Center (BZKF), Munich, Germany. ✉E-mail: faessler@biochem.mpg.de

showed in an independent study that USP9X can deubiquitinate the α5-subunit (Itga5) in cells upon treatment with soluble FN (Kharitidi et al, 2015). Importantly, however, tissues contain primarily FN that is crosslinked by the lysyl oxidase into an insoluble fibrillar network (Melamed et al, 2023) which, in contrast to soluble FN, cannot be internalized by integrins, raising the question whether USP9X also controls the steady state levels of unbound α5β1 integrins.

In the present paper, we designed unbiased genetic and biochemical screens aimed at identifying novel DUB(s) that maintain Itgb1 levels at the cell surface at a steady state. Our experiments revealed that the DUBs USP12 and USP46 complexed with WDR48 and WDR20 remove ubiquitin from the cytoplasmic tails of internalized Itgb1 and several other cells surface proteins including signaling proteins and solute transporters, resulting in a decoupling from ESCRT-mediated degradation. The significance of our findings is discussed.

## Results

### The Itgb1 protein is stabilized by USP12 and USP46

As USP9X was shown to deubiquitinate Itga5 tails on endosomes following soluble FN stimulation (Kharitidi et al, 2015), we first investigated whether USP9X also deubiquitinates and stabilizes Itgb1 in cells cultured under steady-state conditions. To this end, we cultured USP9X-depleted mouse fibroblasts, Hela, RPE-1, and MDA-MB-231 cells, respectively, either continuously in the presence of fetal bovine serum (10%, high concentration of soluble serum FN) or in the presence of serum-replacement medium (which lacks soluble FN). The experiments revealed that depletion of USP9X in all cells analysed was either without effect or slightly increased rather than decreased Itga5 and Itgb1 levels on the surface which was measured by flow cytometry, and in lysates which was determined by Western blot (WB), irrespective of whether exposed to medium containing or lacking FBS (Fig. EV1A–H). These findings indicate that USP9X does not control integrin turnover under steady state culture conditions and suggest that unknown DUB(s) ensure retrieval of integrins.

These results prompted us to design an unbiased Crispr/Cas9-based genetic screen aimed at identifying DUBs that regulate the surface stability of Itgb1 in cells cultured with 10% FBS-containing medium at steady state (Fig. 1A). Specifically, we targeted 98 human DUBs genes by transducing the human Cas9-expressing haploid HAP1 cell line with pooled lentiviral guide RNA (gRNA) libraries (Paulmann et al, 2022). The transduced HAP1 cells were then expanded, fixed, immunostained for Itgb1, and sorted by flow cytometry to obtain the 5% cells with the lowest and the 5% cells with the highest Itgb1 surface levels. Next, we used next-generation sequencing (NGS) to identify the gRNA-targeted genes in the Itgb1^low and Itgb1^high cell populations, respectively. We identified *BAP1*, *USP7*, *OTUD6B*, and *USP46* genes in the Itgb1^low, and *PSMD14* and *USP14* in the Itgb1^high as potential regulators of Itgb1 cell surface levels (Fig. 1B).

To identify which of the DUBs identified in the Crispr/Cas9-based screen are present in the proximity of the Itgb1 tail in mouse fibroblasts, we determined the Itgb1 proximitome by combining the proximity-dependent biotin identification (BioID) assay in

combination with mass spectrometry (MS)-based proteomics. First, we fused the miniTurbo (Branon et al, 2018) to the cytosolic tail of Itga5 which associates with Itgb1 whose tail integrity is required to bind interactors such as Kindlins and SNX17 (Böttcher et al, 2012; Fitzpatrick et al, 2014; Li et al, 2017). The Itga5-miniTurbo was retrovirally transduced into wild-type (WT) and Itgb1-KO fibroblasts. The newly synthesized Itga5 cannot heterodimerize in the absence of Itgb1 and is degraded in the endoplasmatic reticulum, which makes Itgb1-KO cells a perfect negative control. Next, we isolated biotinylated proteins from cell lysates with streptavidin-conjugated beads, performed MS, and identified USP46, the paralog USP12, and the USP12- and USP46-binding and activating adapter proteins WDR48 and WDR20 (Li et al, 2016; Zhu et al, 2019) (Fig. 1C). USP12 and USP46 share approximately 90% protein sequence similarity and conserved binding sites for WDR48 and WDR20 (Li et al, 2016; Zhu et al, 2019). The BAP1, USP7, OTUD6B, and USP14 proteins were undetectable by MS. PSMD14 and USP9X were detected at comparably low levels in WT and Itgb1-KO fibroblasts, suggesting that these two proteins exhibit background binding, e.g., to the beads used in the experiment. Thus, the unbiased genetic screen as well as the proximitome point to the USP12/46-WDR48-WDR20 DUB complex (hereafter referred to as USP12/46-WDRs complex) as a stabilizer of the Itgb1 surface levels. Interestingly, *USP12* found in the proximity of Itga5, was not identified in our Crispr/Cas9-based screen. It is possible that the low expression of USP12 in HAP1 cells (according to the Human Protein Atlas USP46 levels are almost twofold higher than USP12 levels) and/or the incomplete Crispr/Cas9-mediated depletion were responsible that USP12 escaped detection in the Crispr/Cas9 screening.

To validate the results of our screens, we used Crispr/Cas9 technology to knockout (KO) the *USP12*, *USP46*, *WDR20*, and *WDR48* genes either individually or in combination in at least two different mouse fibroblast and human breast cancer MDA-MB-231 cell clones, respectively. Since antibodies against the USP12/46-WDRs complex are not available and several attempts to generate specific homemade polyclonal antisera were unsuccessful, we validated the KOs of the individual clones by genomic PCR followed by sequencing of the amplified genes (Fig. EV2A–G). Flow cytometry analysis revealed that the expression levels of Itgb1 on fibroblasts carrying single KO of either USP12 or USP46 were comparable to those of wild-type fibroblasts (Fig. 1D), whereas the levels of Itgb1 on fibroblasts with the double knockout of USP12 and USP46 (USP12/46-dKO) were significantly reduced (Fig. 1D), suggesting that USP12 and USP46 compensate each other. WB of fibroblast lysates revealed that the levels of the 105 kDa Itgb1 band corresponding to the immature, ER-resident Itgb1 remained unaffected by the USP12/46-dKO, while the levels corresponding to the 125 kDa mature Itgb1 band were reduced in USP12/46-dKO cells (Fig. 1E,F), indicating that the destabilization occurs either in the secretory pathway, on the cell surface and/or in the endosomal system. Concomitantly with the decrease of the mature Itgb1 also the Itga5 levels were reduced in USP12/USP46-dKO lysates (Fig. 1E,F). In MDA-MB-231 cells, the dKO of USP12/46 also reduced the levels of Itgb1 at the cell surface, and those of the 125 kDa mature Itgb1 in the whole cell lysate (Fig. EV3A–C).

The reduced Itgb1 levels in USP12/46-dKO fibroblasts were restored upon expression of either EGFP-tagged USP12^WT or Flag-tagged USP46^WT (Figs. 1G–I and EV3D–G). In contrast,

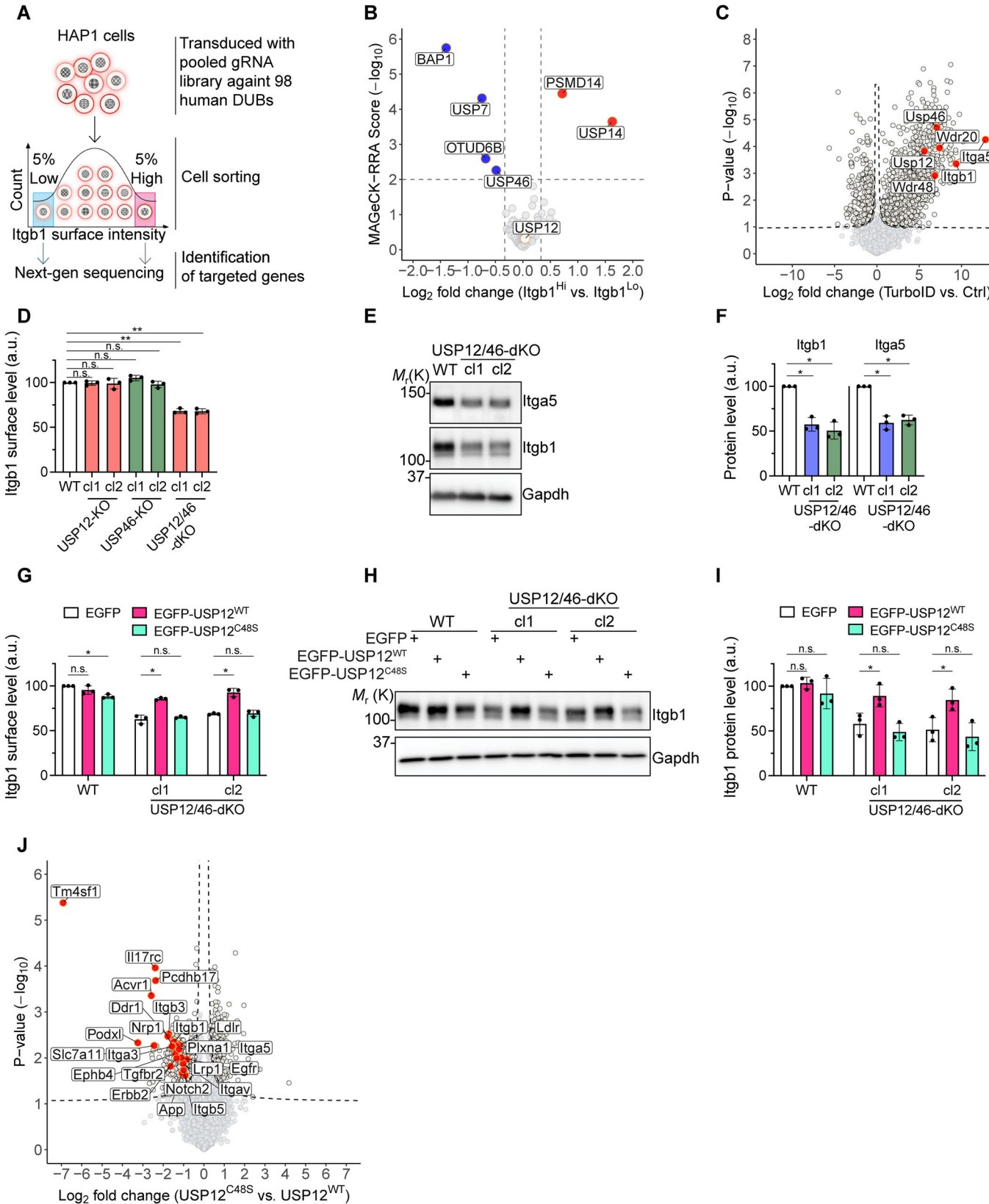

**Figure 1.  The USP12/46-WDRs complex maintains Itgb1 protein levels.**

(A) Schematic overview of the CRISPR screen for identifying DUBs regulating Itgb1 surface levels. Cas9-expressing HAP1 cells were transduced with pooled lentiviral guide RNA (gRNA) libraries targeting 98 DUBs from the human genome. After 2 weeks in culture, cells with the 5% lowest (Itgb1$^{Lo}$) and the 5% highest (Itgb1$^{Hi}$) Itgb1 surface levels were sorted by flow cytometry, and gRNA-targeted genes were determined. (B) Volcano plot of the results from the CRISPR screen. The x-axis represents the log$_2$ fold change (lfc) in the frequency of genes targeted between the Itgb1$^{Lo}$ and Itgb1$^{Hi}$ populations. The y-axis indicates the robust rank aggregation (RRA) score determined by the MAGeCK algorithm (MAGeCK-RRA) (Li et al, 2014). Dots represent individual targeted genes, and those meeting the criteria of |lfc| >0.33 and $-\log_{10}$(RRA) >2 were considered significant. Genes significantly enriched in Itgb1$^{Lo}$ cells are colored in blue, and those enriched in Itgb1$^{Hi}$ in red. USP12 is marked in white. (C) Volcano plot of the α5β1 integrin proximitome determined by label-free MS analysis in mouse kidney fibroblasts expressing miniTurbo-tagged Itag5 (TurboID) versus Itgb1-KO fibroblasts (Ctrl). $P$ values were determined using a two-sided permuted $t$-test with 250 randomizations. The black dashed line indicates the significance cutoff (FDR:0.05, S0:0.1) estimated by the Perseus software. $n = 3$ biological replicates. The red dots indicate the subunits of the α5β1 heterodimer and the components of the USP12/46-WDR48-WDR20 complex. (D) Itgb1 surface levels in WT and two independent clones (cl1 and cl2) of USP12-KO, USP46-KO, and USP12/46-dKO fibroblasts determined by flow cytometry. Statistical analysis was carried out by RM one-way ANOVA with Dunnett's multiple comparison test comparing the WT fibroblasts with USP12-KO cl1 or cl2, USP46-KO cl1 or cl2, USP12/46-KO cl1 or cl2 fibroblasts ($P = 0.9905, 0.9968, 0.2401, 0.8080, 0.0067,$ and $0.0065$, respectively). **$P < 0.01$; n.s. not significant. Data were shown as Mean ± SD, $n = 3$ independent experiments. (E, F) WB (E) and densitometric quantification (F) of Itgb1 and Itga5 protein levels in WT and USP12/46-dKO fibroblasts. Gapdh served as loading control. Statistical analysis was carried out by RM one-way ANOVA with Dunnett's multiple comparison test comparing the WT fibroblasts with USP12/46-KO cl1 or cl2 fibroblasts (for Itgb1, $P = 0.0162$ and $0.0189$, respectively; for Itga5, $P = 0.0180$ and $0.0106$, respectively). *$P < 0.05$. Data were shown as Mean ± SD, $n = 3$ independent experiments. (G–I) Itgb1 surface levels were determined by flow cytometry (G), Itgb1 protein levels in cell lysates were determined by WB (H), and densitometric quantification (I) in WT and USP12/46-dKO fibroblasts stably expressing EGFP, EGFP-USP12$^{WT}$, or EGFP-USP12$^{C48S}$. Gapdh served as a loading control. Statistical analysis was carried out by RM two-way ANOVA with Dunnett's multiple comparison test. In (G), statistical significance was tested comparing the EGFP group with EGFP-USP12$^{WT}$ or EGFP-USP12$^{C48S}$ group in WT fibroblasts ($P = 0.3449$ and $0.0193$, respectively); in USP12/46-dKO cl1 fibroblasts ($P = 0.0335$ and $0.6230$, respectively); and in USP12/46-dKO cl2 fibroblasts ($P = 0.0184$ and $0.9260$, respectively). In (I), statistical significance was tested comparing the EGFP group with EGFP-USP12$^{WT}$ or EGFP-USP12$^{C48S}$ group in WT fibroblasts ($P = 0.6467$ and $0.6716$, respectively); in USP12/46-dKO cl1 fibroblasts ($P = 0.0266$ and $0.3351$, respectively); and in USP12/46-dKO cl2 fibroblasts ($P = 0.0332$ and $0.1310$, respectively). *$P < 0.05$; n.s. not significant. Data were shown as Mean ± SD, $n = 3$ independent experiments. (J) Volcano plot of the cell surface proteome of USP12/46-dKO fibroblasts stably expressing EGFP-USP12$^{C48S}$ versus EGFP-USP12$^{WT}$ identified by label-free MS. $P$ values were determined using two-sided permuted $t$-test with 250 randomizations. The black dashed line indicates the significance cutoff (FDR:0.05, S0:0.1) estimated by the Perseus software. $n = 4$ biological replicates. Arbitrarily selected cell surface receptors were highlighted in red. Source data are available online for this figure.

re-expression of the catalytically inactive USP12$^{C48S}$ or USP46$^{C44S}$ mutants, in which the catalytic site cysteine was substituted for serine (Li et al, 2016; Yin et al, 2015), were unable to restore the Itgb1 levels indicating that USP12/46 require the DUB activity to maintain the 125 kDa mature Itgb1 levels (Figs. 1G–I and EV3D–G). Since USP12 and USP46 compensate each other, we used USP12 to delineate the DUB function in the following reconstitution experiments.

To assess whether USP12 regulates surface proteins other than Itgb1, we determined the cell surface proteome of USP12/46-dKO fibroblasts (Fig. 1J) and USP12/46-dKO MDA-MB-231 cells reconstituted with either USP12$^{WT}$ or USP12$^{C48S}$ (Fig. EV3H). To this end, we biotinylated cell surface proteins, precipitated the biotinylated proteins using streptavidin-conjugated beads, and compared the abundance of the precipitated proteins by MS. We found that the levels of numerous surface receptors, including integrins (Itgb3, Itgb5, Itga3, and Itgav), IL17rc, Pcdhb17, Acvr1, Ddr1, etc. were significantly decreased in USP12$^{C48S}$ expressing fibroblasts (Fig. 1J). Decreased surface levels of integrins, FAT4, STEAP3, PLXNB3, FZD6, etc. were identified in USP12$^{C48S}$ expressing MDA-MB-231 cells (Fig. EV3H). These results indicate that USP12 controls the levels of numerous surface proteins.

## Binding of USP12 to WDR48-WDR20 is essential to maintain Itgb1 surface levels

Previous studies have shown that the deubiquitinase activity of USP12 and USP46 requires the association with the adapter proteins WDR48 or WDR20 and is further increased upon binding to both, WDR48 and WDR20 (Li et al, 2016; Zhu et al, 2019). In line with these findings, we observed that Crispr/Cas9-mediated KO of either WDR48 or WDR20 moderately decreased Itgb1 surface levels on independently generated fibroblast clones, whereas the dKO of WDR48, as well as WDR20, decreased

Itgb1 surface levels to the same extent as in USP12/46-dKO or USP12$^{C48S}$-expressing USP12/46-dKO fibroblasts (Fig. 2A). Furthermore, expression of either WDR48 or WDR20 alone in WDR48/20-dKO fibroblasts restored the levels of Itgb1 at the cell surface to a lesser extent than expression of the two WDR proteins together (Fig. 2B).

We also confirmed that the activity of USP12 depends on the direct interaction with WDR48 and WDR20 (Dharadhar et al, 2016; Li et al, 2016) by mutating the binding site in EGFP-tagged USP12 for WDR48 (USP12$^{1XMUT}$, E190K) (Dharadhar et al, 2016), for WDR20 (USP12$^{2XMUT}$, F287A, V279A) (Li et al, 2016) or for both, WDR48 and WDR20 (USP12$^{3XMUT}$, E190K, F287A, V279A) (Fig. EV4A). Expression of USP12$^{WT}$ in USP12/46-dKO fibroblasts rescued Itgb1 surface levels, whereas expression of USP12$^{1XMUT}$ or USP12$^{2XMUT}$ only partially rescued Itgb1 surface levels and expression of USP12$^{3XMUT}$ did not increase Itgb1 levels beyond the levels of USP12/46-dKO fibroblasts expressing EGFP-only (Figs. 2C and EV4B). Also the expression of EGFP-tagged WDR48 mutant proteins (WDR48$^{MUT}$, K214E/W256A/R272D) in WDR48-KO fibroblasts, and WDR20 mutant proteins (WDR20$^{MUT}$, F262A/W306A) in WDR20-KO fibroblasts, both of which are unable to bind USP12/46 (Li et al, 2016; Yin et al, 2015), failed to normalize Itgb1 surface levels (Figs. 2D,E and EV4C–F). These findings indicate that the entire USP12/46-WDRs complex is required to stabilize Itgb1.

The WDR48 protein consists of an N-terminal propeller domain followed by an ancillary domain (AD) and a C-terminal sumo-like domain (SLD), which is thought to recruit the substrate (in our case, the ubiquitinated Itgb1 tail) to the USP-WDR48 complex (Li et al, 2016; Yin et al, 2015). However, expression of EGFP-tagged WDR48 protein lacking the SLD (WDR48$^{1-580}$) or the SLD as well as the AD (WDR48$^{1-359}$) domain in WDR48-KO fibroblasts restored Itgb1 surface levels to the same extent as expression of WDR48$^{WT}$, indicating that neither the SLD nor the AD domains are required to control DUB-mediated Itgb1 surface levels (Fig. EV4G–I).

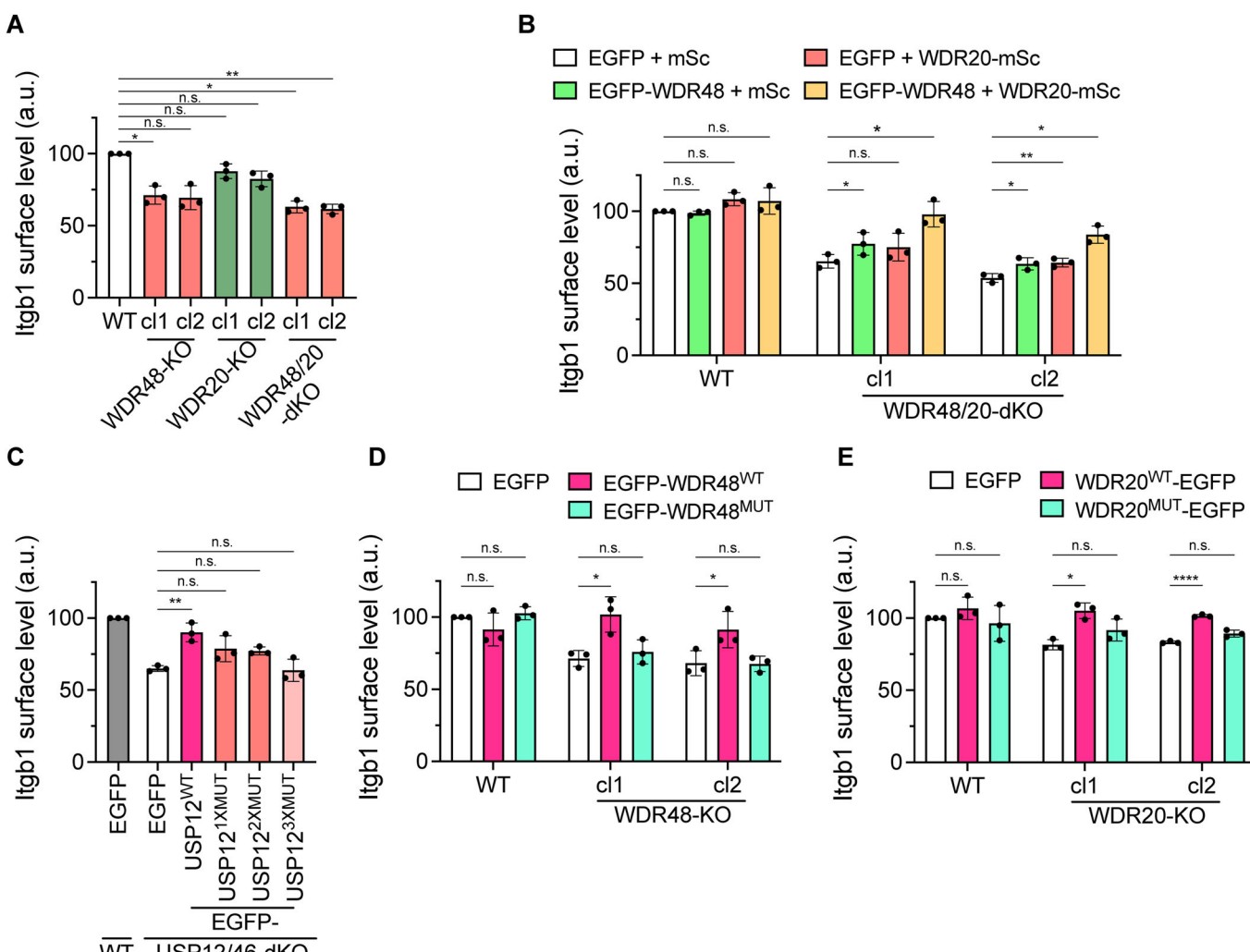

**Figure 2. The integrity of the USP12/46-WDR20-WDR48 complex is required to maintain Itgb1 protein levels.**

(A) Itgb1 surface levels in WT and WDR48-KO, WDR20-KO, and WDR48/20-dKO fibroblasts were determined by flow cytometry. Statistical analysis was carried out by RM one-way ANOVA with Dunnett's multiple comparison test comparing the WT fibroblasts with WDR48-KO cl1 or cl2, WDR20-KO cl1 or cl2, WDR48/20-dKO cl1 or cl2 fibroblasts ($P = 0.0417$, 0.0629, 0.1347, 0.0788, 0.0113 and 0.0065, respectively). *$P < 0.05$; **$P < 0.01$; n.s. not significant. Data were shown as Mean ± SD, $n = 3$ independent experiments. (B) Itgb1 surface levels in WT and WDR48/20-dKO fibroblasts transiently expressing EGFP, mScarlet, EGFP-WDR48, and/or WDR20-mScarlet determined by flow cytometry. Statistical analysis was carried out by RM two-way ANOVA with Dunnett's multiple comparison test comparing the EGFP + mSc group with EGFP-WDR48 + mSc group, EGFP + WDR20-mSc group or EGFP-WDR48 + WDR20-mSc group in WT fibroblasts ($P = 0.4902$, 0.1622, and 0.5335, respectively); in WDR48/20-dKO cl1 fibroblasts ($P = 0.0432$, 0.2189, and 0.0178, respectively); and in WDR48/20-dKO cl2 fibroblasts ($P = 0.0173$, 0.0036, and 0.0305, respectively). *$P < 0.05$; **$P < 0.01$; n.s. not significant. Data were shown as Mean ± SD, $n = 3$ independent experiments. (C) Itgb1 surface levels in WT and USP12/46-dKO fibroblasts transiently expressing EGFP, EGFP-USP12WT, EGFP-USP12 1XMUT (E190K, deficient in binding to WDR48), EGFP-USP12 2XMUT (F287A, V279A, deficient in binding to WDR20), or EGFP-USP12 3XMUT (E190K, F287A, V279A, deficient in binding to WDR48 as well as WDR20) determined by flow cytometry. Statistical analysis was carried out by ordinary one-way ANOVA with Dunnett's multiple comparison test comparing the EGFP group with EGFP-USP12WT, EGFP-USP12 1XMUT, EGFP-USP12 2XMUT, or EGFP-USP12 3XMUT group in USP12/46-dKO fibroblasts ($P = 0.0021$, 0.0680, 0.1116, and 0.9972, respectively). **$P < 0.01$; n.s. not significant. Data were shown as Mean ± SD, $n = 3$ independent experiments. (D) Itgb1 surface levels in WT and WDR48-KO fibroblasts transiently expressing EGFP-WDR48WT or EGFP-WDR48MUT (deficient in binding to USP12 as well as USP46) determined by flow cytometry. Statistical analysis was carried out by RM two-way ANOVA with Dunnett's multiple comparison test comparing EGFP group with EGFP-WDR48WT or EGFP-WDR48MUT group in WT fibroblasts ($P = 0.4661$ and 0.5964, respectively); in WDR48-KO cl1 fibroblasts ($P = 0.0281$ and 0.4168, respectively); and in WDR48-KO cl2 fibroblasts ($P = 0.0154$ and 0.9882, respectively). *$P < 0.05$; n.s. not significant. Data were shown as Mean ± SD, $n = 3$ independent experiments. (E) Itgb1 surface levels in WT and WDR20-KO fibroblasts transiently expressing WDR20WT-EGFP or WDR20MUT-EGFP (deficient in binding to USP12 as well as USP46) determined by flow cytometry. Statistical analysis was carried out by RM two-way ANOVA with Dunnett's multiple comparison test comparing the EGFP group with WDR20WT-EGFP or WDR20MUT-EGFP group in WT fibroblasts ($P = 0.4034$ and 0.8410, respectively); in WDR20-KO cl1 fibroblasts ($P = 0.0144$ and 0.2935, respectively); and in WDR20-KO cl2 fibroblasts ($P < 0.0001$ and $P = 0.1234$, respectively). *$P < 0.05$; ****$P < 0.0001$; n.s. not significant. Data were shown as Mean ± SD, $n = 3$ independent experiments. Source data are available online for this figure.

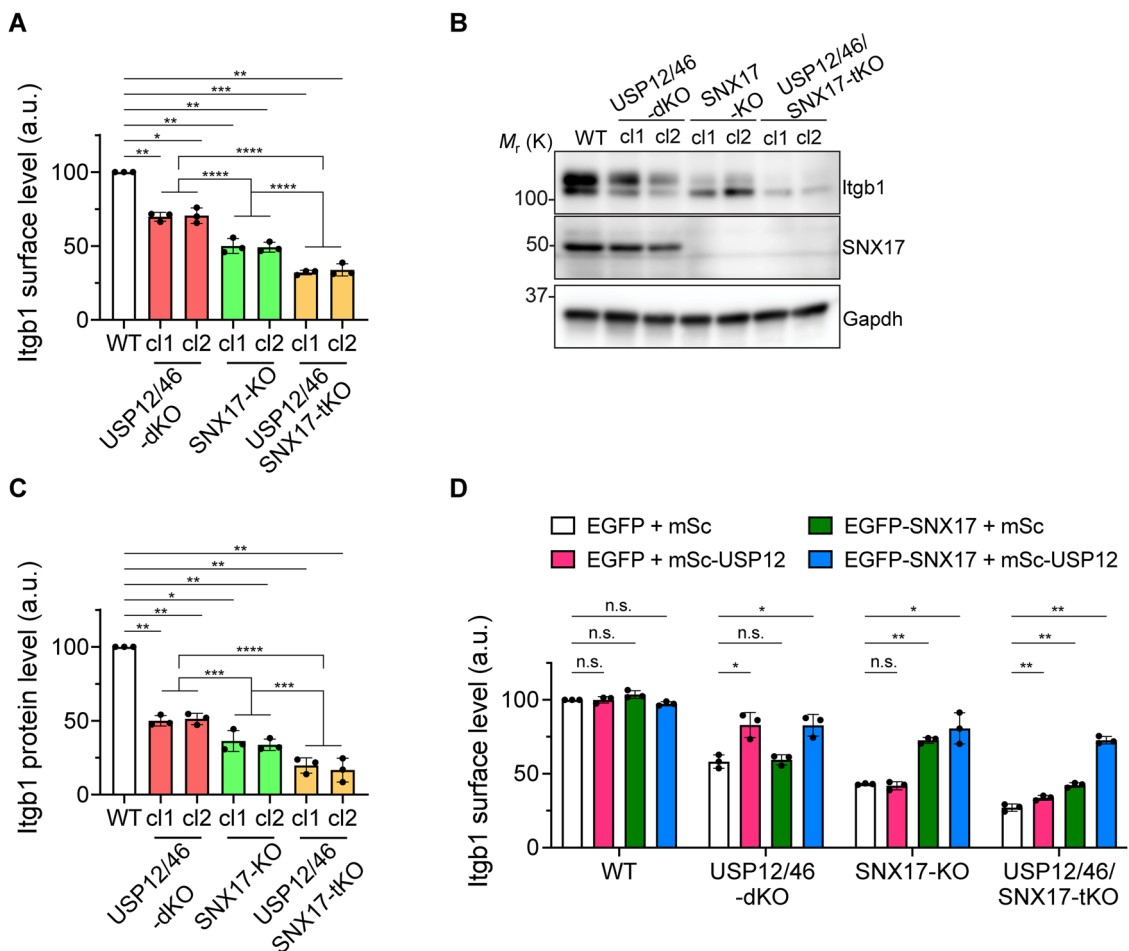

**Figure 3.  The USP12/46-WDRs complex maintains Itgb1 protein levels in a SNX17-independent manner.**

(A–C) Itgb1 surface levels were determined by flow cytometry (A), and Itgb1 and SNX17 protein levels in cell lysates were determined by WB (B) with densitometric quantification (C) in WT, USP12/46-dKO, SNX17-KO, and USP12/46/SNX17-tKO fibroblasts. Gapdh served as a loading control. In (A), statistical analysis was carried out by RM one-way ANOVA with Dunnett's multiple comparison test comparing the WT fibroblasts with USP12/46-dKO cl1 or cl2, SNX17-KO cl1 or cl2, USP12/46/SNX17-tKO cl1 or cl2 fibroblasts ($P = 0.0086, 0.0273, 0.0088, 0.0038, 0.0005,$ and $0.0035,$ respectively). Then ordinary one-way ANOVA with Tukey's multiple comparison test was used to make comparisons between USP12/46-dKO and SNX17-KO fibroblasts ($P < 0.0001$); between USP12/46-dKO and USP12/46/SNX17-tKO fibroblasts ($P < 0.0001$) and between SNX17-KO and USP12/46/SNX17-tKO fibroblasts ($P < 0.0001$). In (C), statistical analysis was carried out by RM one-way ANOVA with Dunnett's multiple comparison test comparing the WT fibroblasts with USP12/46-dKO cl1 or cl2, SNX17-KO cl1 or cl2, USP12/46/SNX17-tKO cl1 or cl2 fibroblasts ($P = 0.0044, 0.0054, 0.0109, 0.0031, 00037,$ and $0.0082,$ respectively). Then ordinary one-way ANOVA with Tukey's multiple comparison test was used to make comparisons between USP12/46-dKO and SNX17-KO fibroblasts ($P = 0.0003$); between USP12/46-dKO and USP12/46/SNX17-tKO fibroblasts ($P < 0.0001$) and between SNX17-KO and USP12/46/SNX17-tKO fibroblasts ($P = 0.0001$). *$P < 0.05$; **$P < 0.01$; ***$P < 0.001$; ****$P < 0.0001$. Data were shown as Mean ± SD, $n = 3$ independent experiments. (D) Itgb1 surface levels in WT, USP12/46-dKO, SNX17-KO, and USP12/46/SNX17-tKO fibroblasts transiently expressing indicated combinations of constructs determined by flow cytometry. Statistical analysis was carried out by RM two-way ANOVA with Dunnett's multiple comparison test comparing the EGFP + mSc group with EGFP + mSc-USP12 group, EGFP-SNX17 + mSc group or EGFP-SNX17 + mSc-USP12 group in WT fibroblasts ($P > 0.9999, P = 0.2554$ and $0.1691,$ respectively); in USP12/46-dKO fibroblasts ($P = 0.0210, 0.6588$ and $0.0147,$ respectively); in SNX17-KO fibroblasts ($P = 0.8238, 0.0031$ and $0.0460,$ respectively); and in USP12/46/SNX17-tKO fibroblasts ($P = 0.0099, 0.0058$ and $0.0069,$ respectively). *$P < 0.05$; **$P < 0.01$; n.s. not significant. Data were shown as Mean ± SD, $n = 3$ independent experiments. Source data are available online for this figure.

## SNX17 and the USP12/46-WDRs complex stabilize Itgb1 independently of each other

SNX17 binds Itgb1 on early endosomes and was shown to promote the recycling of Itgb1 in an Itgb1-tail ubiquitination-dependent manner (Böttcher et al, 2012; Steinberg et al, 2012). Since dKO of USP12/46 reduced Itgb1 cell surface levels and total levels in cell lysates to a similar extent as loss of SNX17, we tested whether the deletion of the *Snx17* gene in USP12/46-dKO fibroblasts affects Itgb1 surface levels. The experiment revealed that the cell surface

and total Itgb1 levels in USP12/46/SNX17-triple (t)KO fibroblast clones were further decreased compared to those in the USP12/46-dKO fibroblasts, suggesting that the DUB complex and SNX17 act independently of each other in maintaining Itgb1 levels (Fig. 3A–C). Re-expression of USP12 in USP12/46-dKO or SNX17 in SNX17-KO fully restored the surface levels of Itgb1 (Fig. 3D). Co-expression of USP12 and SNX17 fully restored the Itgb1 levels in tKO fibroblasts, whereas separate expression of USP12 or SNX17 in tKO fibroblasts failed to normalize Itgb1 surface levels (Fig. 3D).

## The USP12/46-WDRs complex prevents lysosomal degradation of Itgb1

Next, we investigated the mechanism underlying the downregulation of Itgb1 expression in USP12/46-dKO cells. A role for *Itgb1* mRNA transcript stability in regulating Itgb1 protein levels could be excluded as *Itgb1* mRNA levels did not decrease in the USP12/46-dKO fibroblasts (Fig. 4A). Cycloheximide (CHX) chase assays, which allow to compare the degradation kinetics of proteins, revealed accelerated Itgb1 protein degradation in USP12/46-dKO compared to WT fibroblasts (Fig. 4B,C). Furthermore, surface biotinylation followed by capture-ELISA (Böttcher et al, 2012) showed a significantly reduced Itgb1 surface stability in USP12/46-dKO fibroblasts compared to WT fibroblasts. The Itgb1 protein stability was much lower in USP12/46-dKO fibroblasts than in WT fibroblasts (Fig. 4D). The reduced Itgb1 surface stability in USP12/46-dKO fibroblasts was restored upon re-expression of USP12^WT but not upon re-expression of the catalytically inactive USP12^C48S (Fig. 4E). We also found that the internalization kinetics of Itgb1 was similar between USP12^WT and USP12^C48S re-expressing USP12/46-dKO fibroblasts (Fig. 4F), whereas the recycling rate of Itgb1 was reduced in USP12^C48S expressing fibroblasts compared to USP12^WT expressing fibroblasts (Fig. 4G). These data indicate that the catalytic activity of the USP12/46-WDRs complex maintains the surface as well as total Itgb1 protein levels by enabling the recycling of internalized Itgb1 to the cell surface.

To determine the pathway through which Itgb1 is degraded in the absence of USP12/46, we treated USP12/46-dKO fibroblasts with MG132 or Bafilomycin A1 (BafA1). Whereas inhibition of the proteasome with MG132 did not restore Itgb1 levels, inhibition of the lysosome with BafA1 restored total Itgb1 levels in cell lysates (Fig. 4H,I). Surprisingly, however, BafA1 treatment did not restore the surface levels of Itgb1 in USP12/46-dKO fibroblasts (Fig. 4J), while BafA1 treatment normalized Itgb1 surface levels in SNX17-KO cells (Böttcher et al, 2012).

To determine the subcellular localization of Itgb1 in WT and USP12/46-dKO fibroblasts, we immunostained fixed cells and found Itgb1 in FAs, ER, and a few intracellular puncta in both, WT and USP12/46-dKO fibroblasts. Following BafA1 treatment, we observed large Itgb1-positive puncta co-stained with the late endosome/lysosome marker Lamp1 in USP12/46-dKO fibroblasts, which were rarely observed in WT fibroblasts (Fig. 4K). An increased Pearson correlation coefficient (PCC) of Itgb1 with Lamp1 in USP12/46-dKO fibroblasts confirmed the increased endo/lysosomal localization of Itgb1 in USP12/46-dKO compared to WT fibroblasts (Fig. 4L). Collectively, these findings indicate that USP12/46 depletion leads to increased lysosomal targeting of Itgb1, resulting in degradation, impaired recycling, and reduced Itgb1 surface levels.

## The USP12/46-WDRs complex prevents ESCRT-mediated degradation of Itgb1

Since BafA1 prevents the degradation of proteins in lysosomes by blocking the vacuolar-type ATPase and the lysosomal acidification (Wang et al, 2021), we hypothesized that the absent Itgb1 deubiquitination in USP12/46-dKO cells promotes ESCRT binding, internalization of Itgb1 via intraluminal vesicles (ILVs) and Itgb1 degradation. To test this hypothesis, we conjointly depleted ESCRT-0 component HGS and ESCRT-I component TSG101 by

siRNAs (ESCRT-KD) in WT and USP12/46-dKO fibroblasts and found that Itgb1 levels and stability increased in both, WT^ESCRT-KD and USP12/46-dKO^ESCRT-KD fibroblasts (Fig. 5A–D). Immunoprecipitation of endogenous Itgb1 followed by WB for ubiquitinated Itgb1 with anti-Ub antibody revealed elevated poly-ubiquitination of Itgb1 in USP12/46-dKO^ESCRT-KD fibroblasts compared to WT^ESCRT-KD fibroblasts (see smears between 130–200 kDa in Fig. 5E,F). The increased Itgb1 ubiquitination in USP12/46-dKO^ESCRT-KD fibroblasts was confirmed by capturing ubiquitinated proteins using ubiquitin-trap beads and subsequently probing the gel-separated proteins with a polyclonal anti-Itgb1 antibody (Fig. 5G,H).

To identify the lysine residues in the Itgb1 tail that are ubiquitinated, we immunoprecipitated ubiquitinated proteins, treated the precipitate with trypsin, and used MS to identify peptides containing a Gly-Gly motif, which is a remaining signature of an Ub-conjugated site (Xu et al, 2010). The experiment identified lysine-774 (K774), K784 and K794 modified by ubiquitin (Fig. 5I; Table EV1). Interestingly, K794 is located within the Kindlin- and SNX17-binding NPK$_{794}$Y motif, and K784 is adjacent to the Talin-binding NPIY$_{783}$ motif.

The ubiquitin (Ub) chain linkage specificity determines the fate and function of polyubiquitinated proteins (Miranda and Sorkin, 2007; Swatek and Komander, 2016). Lysine-63 (K63)-linked polyUb chains are preferentially associated with ESCRT-mediated lysosomal degradation, while lysine-48 (K48)-linked chains lead to proteasomal degradation (Nathan et al, 2013; Strickland et al, 2022). To inhibit K48- or K63-mediated Ub conjugation, we overexpressed ubiquitin mutants carrying K48R or K63R substitutions (Lim et al, 2005) and concomitantly depleted WT and USP12/46-dKO fibroblasts with ESCRT-KD siRNAs to enrich for ubiquitinated Itgb1 (Fig. 5J,K). The experiment revealed an increase of ubiquitinated Itgb1 in cells expressing Ub^WT or Ub^K48R, but not Ub^K63R. The levels of Itgb1 ubiquitinated with Ub^WT and Ub^K48R were higher in USP12/46-dKO^ESCRT-KD compared to WT^ESCRT-KD fibroblasts, which indicates that K63-mediated polyUb chain modification destines Itgb1 for lysosomal degradation.

To further confirm the role of the USP12/46-WDRs complex in the deubiquitination of Itgb1, we performed an in vitro deubiquitination assay with recombinantly produced ternary wild-type USP12^WT-WDR48-WDR20 or catalytically inactive USP12^C48S-WDR48-WDR20 complexes (Fig. 6A,B). To enrich for ubiquitinated Itgb1, fibroblasts were first depleted with ESCRT-KD siRNAs, and then Itgb1 was immunoprecipitated from cell lysates and incubated with the recombinant USP12^WT-WDRs or the recombinant USP12^C48S-WDRs complex (Fig. 6C). Whereas USP12^WT-WDRs reduced Itgb1 ubiquitination, the catalytically inactive USP12^C48S-WDRs did not. Furthermore, neither recombinant UCHL5 nor USP7, which are integrin-unrelated DUBs and were used as controls, were able to reduce Itgb1 ubiquitination (Fig. 6A,B). We also found that the EGFP-tagged USP12 expressed in USP12/46-dKO^ESCRT-KD fibroblasts colocalized with Itgb1 and ubiquitin on the limiting membrane of enlarged EEA1-positive endosomes (Fig. 6D–K) and was absent from Itgb1-positive focal adhesions (FAs). Enlarged endosomes were not observed in control siRNA- transfected USP12/46-dKO fibroblasts. Altogether, these data indicate that the USP12/46-WDRs complex colocalizes with Itgb1 on endosomes and deubiquitinates Itgb1, which in turn prevents ESCRT-mediated Itgb1 degradation.

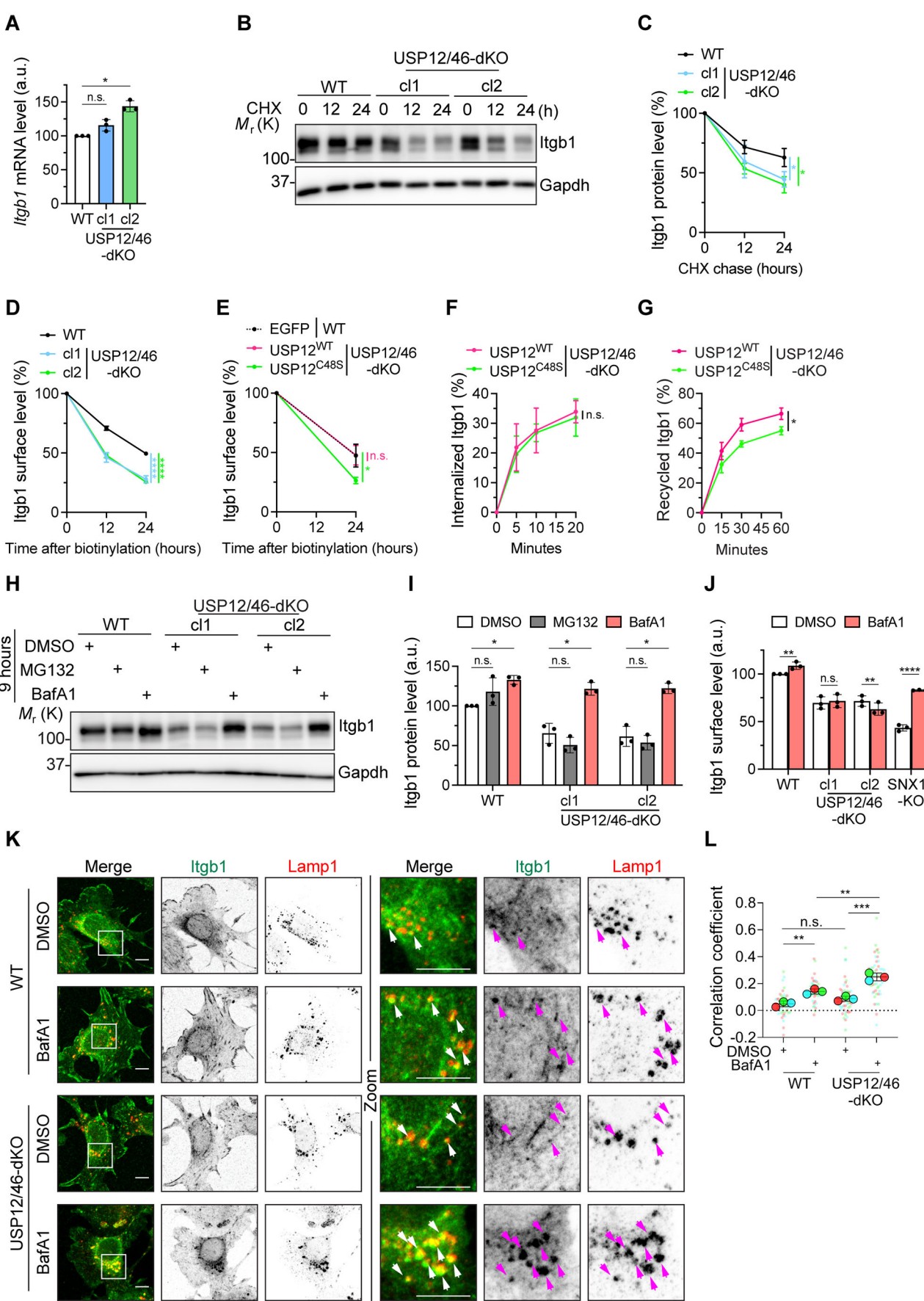

◄  **Figure 4.  USP12/46-WDRs complex prevents lysosomal degradation of Itgb1.**

(A) *Itgb1* mRNA levels in WT and USP12/46-dKO fibroblasts determined by qPCR. Statistical analysis was carried out by RM one-way ANOVA with Dunnett's multiple comparison test comparing the WT fibroblasts with USP12/46-dKO cl1 or cl2 fibroblasts ($P = 0.1278$ and $0.0178$, respectively). *$P < 0.05$; n.s. not significant. Data were shown as Mean ± SD, $n = 3$ independent experiments. (B, C) WB (B) and densitometric quantification (C) of Itgb1 protein levels in WT and USP12/46-dKO fibroblasts at indicated time points after cycloheximide (CHX) treatment (5 µg/ml). Gapdh served as a loading control. Statistical analysis was carried out by ordinary one-way ANOVA with Dunnett's multiple comparison test comparing the WT fibroblasts with USP12/46-dKO cl1 or cl2 fibroblasts at the 24-hour time point ($P = 0.0327$ and $0.0127$, respectively). *$P < 0.05$. Data were shown as Mean ± SD, $n = 3$ independent experiments. (D) Quantification of surface Itgb1 degradation kinetics in WT and USP12/46-dKO fibroblasts. The amount of Itgb1 remaining over indicated times were measured by capture-ELISA. Statistical analysis was carried out by ordinary one-way ANOVA with Dunnett's multiple comparison test comparing the WT fibroblasts with USP12/46-dKO cl1 or cl2 fibroblasts at the 24-h time point (both $P < 0.0001$). ****$P < 0.0001$. Data were shown as Mean ± SD, $n = 3$ independent experiments. (E) Quantification of surface Itgb1 degradation kinetics in WT fibroblasts stably expressing EGFP, and USP12/46-dKO fibroblasts stably expressing EGFP-USP12$^{WT}$ or EGFP-USP12$^{C48S}$ determined by capture-ELISA. Statistical analysis was carried out by ordinary one-way ANOVA with Dunnett's multiple comparison test comparing the WT fibroblasts expressing EGFP with USP12/46-dKO fibroblasts expressing EGFP-USP12$^{WT}$ or EGFP-USP12$^{C48S}$ ($P = 0.9983$ and $0.0248$, respectively). *$P < 0.05$; n.s. not significant. Data were shown as Mean ± SD, $n = 3$ independent experiments. (F, G) The internalization rate (F) and the recycling rate (G) of Itgb1 in USP12/46-dKO fibroblasts stably expressing EGFP-USP12$^{WT}$ or EGFP-USP12$^{C48S}$ determined by capture-ELISA. Statistical analysis was carried out by two-sided Welch's *t*-test comparing USP12/46-dKO fibroblasts expressing EGFP-USP12$^{WT}$ or EGFP-USP12$^{C48S}$ at the end time point. The $P$ values in (F) and (G) are $0.6734$ and $0.0168$, respectively. *$P < 0.05$; n.s. not significant. Data were shown as Mean ± SD, $n = 3$ independent experiments. (H, I) WB (H) with densitometric quantification (I) of Itgb1 protein levels in lysates of WT and USP12/46-dKO fibroblasts treated with DMSO, MG132 (0.5 uM) or BafA1 (40 nM) for 9 h. Gapdh served as a loading control. Statistical analysis was carried out by RM two-way ANOVA with Dunnett's multiple comparison test comparing the DSMO group with MG132 or BafA1 group in WT fibroblasts ($P = 0.3185$ and $0.0151$, respectively); in USP12/46-dKO cl1 fibroblasts ($P = 0.2362$ and $0.0195$, respectively); and in USP12/46-dKO cl2 fibroblasts ($P = 0.1809$ and $0.0344$, respectively). *$P < 0.05$; n.s. not significant. Data were shown as Mean ± SD, $n = 3$ independent experiments. (J) Itgb1 surface levels in WT, USP12/46-dKO, and SNX17-KO fibroblasts treated with DMSO or BafA1 (40 nM) for 9 h determined by flow cytometry. Statistical analysis was carried out by RM two-way ANOVA with Šidák's multiple comparison test comparing DSMO group with BafA1 group in WT fibroblasts ($P = 0.0036$); in USP12/46-dKO cl1 fibroblasts ($P = 0.7237$); in USP12/46-dKO cl2 fibroblasts ($P = 0.0041$); and in SNX17-KO fibroblasts ($P < 0.0001$). **$P < 0.01$; ****$P < 0.0001$; n.s. not significant. Data were shown as Mean ± SD, $n = 3$ independent experiments. (K) Representative immunofluorescence (IF) images of Itgb1 and Lamp1 in WT and USP12/46-dKO fibroblasts treated with DMSO or BafA1 (100 nM) for 3 h. White/pink arrowheads show the accumulation of Itgb1 in Lamp1-positive endo/lysosomes. Boxes indicate magnified cell regions displayed in the Zoom panel. Sum intensity projections from confocal stacks are presented. Scale bar, 10 µm. (L) Superplots showing the Pearson correlation coefficients (PCC) between Itgb1 and Lamp1 in WT and USP12/46-dKO fibroblasts. Statistical analysis was carried out by RM two-way ANOVA with Šidák's multiple comparison test comparing DMSO-treated WT fibroblasts with BafA1-treated WT fibroblasts ($P = 0.0047$); DMSO-treated WT fibroblasts with DMSO-treated USP12/46-dKO fibroblasts ($P = 0.1903$); BafA1-treated WT fibroblasts with BafA1-treated USP12/46-dKO fibroblasts ($P = 0.0019$); DMSO-treated USP12/46-dKO fibroblasts with BafA1-treated USP12/46-dKO fibroblasts ($P = 0.0002$); **$P < 0.01$; ***$P < 0.001$; n.s. not significant. Data were shown as Mean ± SD, $n = 3$ independent experiments; 49 cells were analyzed in DMSO-treated WT cells, 43 in BafA1-treated WT cells, 52 in DMSO-treated USP12/46-dKO cells, and 45 in BafA1-treated USP12/46-dKO cells. Source data are available online for this figure.

Itgb1 crosslinking followed by IP and MS excluded a direct association between Itgb1 and the USP12/46-WDRs complex (Dataset EV1). Since EGFP-USP12 was absent from FAs (Fig. 6D) and the members of the USP12/46-WDRs complex were also undetectable in the integrin adhesome (Horton et al, 2015; Kuo et al, 2011; Schiller et al, 2011), we conclude that the USP12/46-WDRs complex binds Itgb1 on endosomes and not in FAs.

If USP12/46-WDRs complex-mediated deubiquitination prevents Itgb1 degradation, non-ubiquitinated α5β1 integrins in which the lysine residues in the cytoplasmic tails of the α5 and β1 subunits were replaced with arginine residues should also escape ESCRT-mediated degradation and be recycled to the cell surface. To test this hypothesis, we generated Itgb1-KO fibroblasts expressing endogenous USP12/46 (Itgb1-KO$^{WT}$) or lacking USP12/46 (Itgb1-KO$^{USP12/46-dKO}$) and expressed α5$^{WT}$β1$^{WT}$ or α5$^{KR}$β1$^{KR}$, in which the four lysine residues in the Itga5 tail and the eight lysine residues in the Itgb1 tail were substituted for arginine residues (Fig. 7A). As shown previously, surface stabilization of Itgb1 required the substitutions of all lysine residues in the α5- and β1-tails because it could not be achieved by mutating individual or groups of lysines, suggesting that the selectivity for specific lysine residues in the Itgb1-tail is unincisive (Böttcher et al, 2012). BafA1 treatment to block lysosomal degradation followed by immunostaining revealed that α5$^{WT}$β1$^{WT}$ accumulated in Lamp1$^+$ endo/lysosomes in Itgb1-KO$^{USP12/46-dKO}$ fibroblasts, whereas α5$^{KR}$β1$^{KR}$ did not accumulate (Fig. 7B). PCC confirmed the lower colocalization between Lamp1 and α5$^{KR}$β1$^{KR}$ compared to Lamp1 and α5$^{WT}$β1$^{WT}$ (Fig. 7C). Consistent with this observation, BafA1 treatment

increased the surface levels of α5$^{KR}$β1$^{KR}$ and to a much lower extent of α5$^{WT}$β1$^{WT}$ in Itgb1-KO$^{USP12/46-dKO}$ fibroblasts (Fig. 7D). Moreover, treatment with ESCRT-KD siRNAs significantly increased the surface levels of α5$^{WT}$β1$^{WT}$ but barely of α5$^{KR}$β1$^{KR}$ in Itgb1-KO$^{USP12/46-dKO}$ fibroblasts, indicating that the ESCRT binds Ub$^{K63}$ modified Itgb1 tails, which in turn leads to Itgb1 degradation and thereby prevents Itgb1 from being recycled to cell surface (Fig. 7E).

## The stabilization of Itgb1 by USP12/46-WDRs promotes integrin functions

In line with the increased ubiquitination, elevated degradation, and decreased levels of cell surface Itgb1 associated with loss of USP12/46 expression, we observed an impaired adhesion, spreading, and in vitro wound healing-based migration of USP12/46-dKO fibroblasts and transwell migration of USP12/46-dKO MDA-MB-231 cells (Figs. 8A–E and EV5A). Furthermore, USP12/46-dKO MDA-MB-231 cells showed reduced invasion through Matrigel-coated transwell filters compared to WT MDA-MB-231 cells (Fig. 8F,G). In line with these findings, analysis of the TCGC BRCA database linked high expression of USP12 and USP46 with reduced overall survival and progression-free intervals of breast cancer patients (Figs. 8H,I and EV5B–E). These findings indicate that the Itgb1 deubiquitinating and stabilizing function of the USP12/46-WDRs complex is essential for basic integrin functions such as adhesion, spreading, and migration. Furthermore, if their levels surpass a certain threshold, such as in cancer, Itgb1 levels rise, which is advantageous for invading cancer cells.

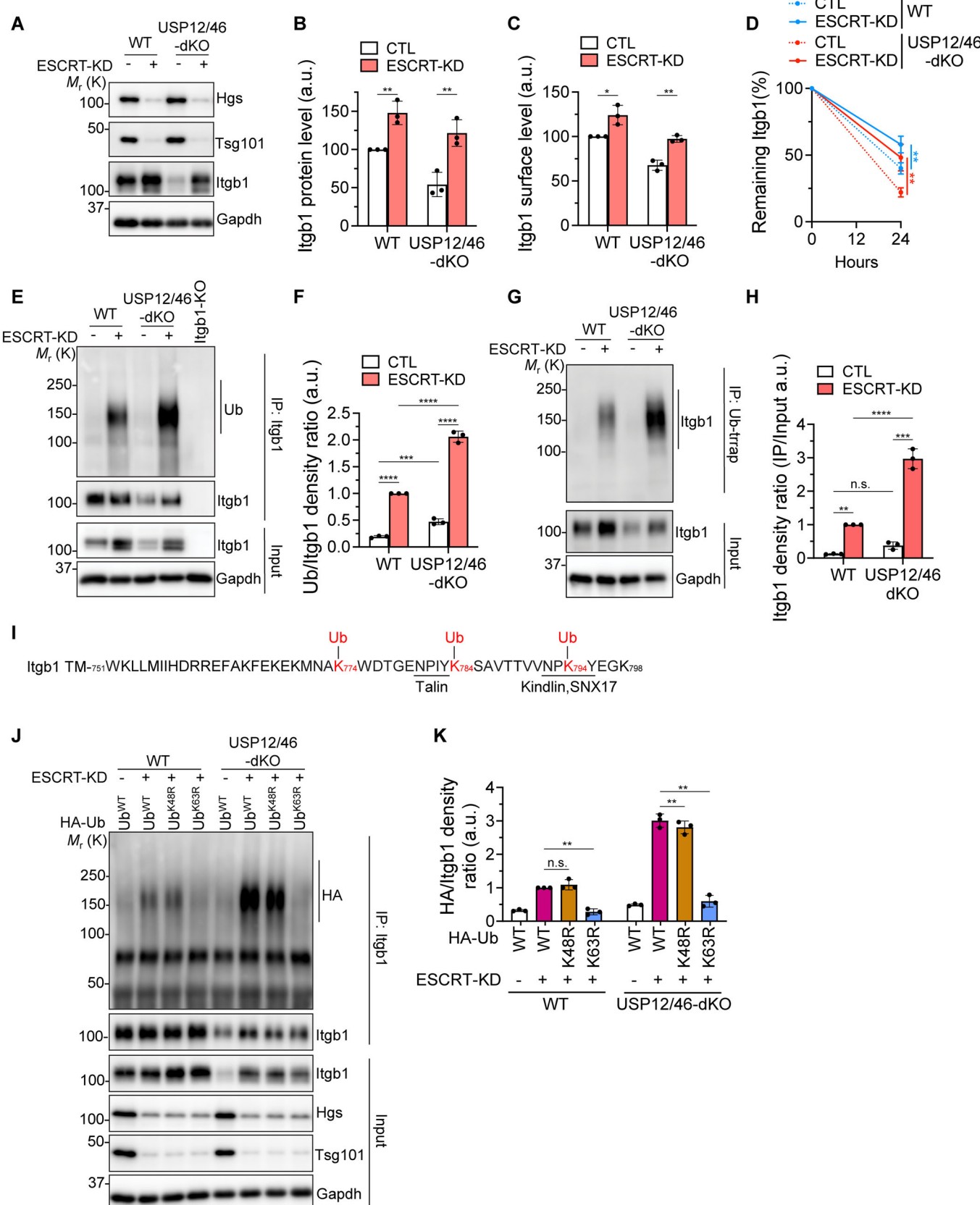

◀ **Figure 5. The USP12/46-WDRs complex deubiquitinates Itgb1 and prevents Itgb1 degradation via the ESCRT pathway.**

(A, B) WB (A) and densitometric quantifications (B) of Itgb1 protein levels in WT and USP12/46-dKO fibroblasts transfected with control non-targeting siRNA (CTL) or siRNAs targeting jointly HGS and TSG101 (ESCRT-KD). Gapdh served as a loading control. Statistical analysis was carried out by RM two-way ANOVA with Šidák's multiple comparison test comparing the CTL group with the ESCRT-KD group in WT fibroblasts ($P = 0.0040$) and in USP12/46-dKO fibroblasts ($P = 0.0011$). **$P < 0.01$. Data were shown as Mean ± SD, $n = 3$ independent experiments. (C) Itgb1 surface levels in WT and USP12/46-dKO fibroblasts treated with or without ESCRT-KD siRNAs determined by flow cytometry. Statistical analysis was carried out by RM two-way ANOVA with Šidák's multiple comparison test comparing the CTL group with the ESCRT-KD group in WT fibroblasts ($P = 0.0132$) and in USP12/46-dKO fibroblasts ($P = 0.0062$). *$P < 0.05$; **$P < 0.01$. Data were shown as Mean ± SD, $n = 3$ independent experiments. (D) Surface Itgb1 degradation kinetics in WT and USP12/46-dKO fibroblasts treated with or without ESCRT-KD siRNAs measured by capture-ELISA. Statistical analysis was carried out by RM two-way ANOVA with Šidák's multiple comparison test comparing the CTL group with the ESCRT-KD group in WT fibroblasts ($P = 0.0064$) and in USP12/46-dKO fibroblasts ($P = 0.0015$). **$P < 0.01$. Data were shown as Mean ± SD, $n = 3$ independent experiments. (E, F) IP of denatured Itgb1 from WT, USP12/46-dKO, and Itgb1-KO fibroblast lysates with or without ESCRT-KD and analyzed by WB (E) for Ubiquitin (Ub) and Itgb1 with quantification (F) of Itgb1 ubiquitination levels. Itgb1-KO served as a negative control. Gapdh served as a loading control. The intensity of the Ub signals was normalized to the intensity of the IP-ed Itgb1 signals. Statistical analysis was carried out by RM two-way ANOVA with Šidák's multiple comparison test comparing the CTL group with ESCRT-KD group in WT fibroblasts ($P < 0.0001$) and in USP12/46-dKO fibroblasts ($P < 0.0001$); comparing CTL groups between WT and USP12/46-dKO fibroblasts ($P = 0.0009$); and ESCRT-KD groups between WT and USP12/46-dKO fibroblasts ($P < 0.0001$). ***$P < 0.001$; ****$P < 0.0001$. Data were shown as Mean ± SD, $n = 3$ independent experiments. (G, H) IP of ubiquitinated proteins in WT and USP12/46-dKO fibroblasts with or without ESCRT-KD and analyzed by WB (G) for Itgb1 with quantification (H) of Itgb1 ubiquitination levels. Gapdh served as a loading control. The intensity of the IP-ed Itgb1 signals was normalized to the intensity of the input Itgb1 signals. Statistical analysis was carried out by RM two-way ANOVA with Šidák's multiple comparison test comparing the CTL group and ESCRT-KD group in WT fibroblasts ($P = 0.0097$) and in USP12/46-dKO fibroblasts ($P = 0.0002$); comparing CTL groups between WT and USP12/46-dKO fibroblasts ($P = 0.1628$); and ESCRT-KD groups between WT and USP12/46-dKO fibroblasts ($P < 0.0001$). **$P < 0.01$; ***$P < 0.001$; ****$P < 0.0001$; n.s. not significant. Data were shown as Mean ± SD, $n = 3$ independent experiments. (I) Amino acid sequence of Itgb1 cytoplasmic tail. The ubiquitin-conjugated lysines detected by MS are colored in red. Data were generated using USP12/46-dKO$^{ESCRT-KD}$ fibroblasts. The underlined regions indicate the NPxY motifs that bind to Talin, Kindlin, and SNX17. See also Table EV1. (J, K) IP of denatured Itgb1 from WT and USP12/46-dKO fibroblasts overexpressing HA-tagged Ub$^{WT}$, Ub$^{K48R}$, or Ub$^{K63R}$ treated with or without ESCRT-KD siRNAs and analyzed by WB (J) for HA and Itgb1 with quantification (K) of Itgb1 ubiquitination levels. Gapdh served as a loading control. The intensity of the HA signals was normalized to the intensity of the IP-ed Itgb1 signals. Statistical analysis was carried out by RM two-way ANOVA with Dunnett's multiple comparison test comparing the Ub$^{WT}$ group with Ub$^{K48R}$ or Ub$^{K63R}$ group in WT fibroblasts ($P = 0.5794$ and 0.0078, respectively) and in USP12/46-dKO fibroblasts ($P = 0.0035$ and 0.0028, respectively). **$P < 0.01$; n.s. not significant. Data were shown as Mean ± SD, $n = 3$ independent experiments. Source data are available online for this figure.

## Discussion

The steady-state surface level of Itgb1 underlies a tightly regulated decision-making process, which routes internalized Itgb1 either to lysosomes for degradation or to recycling endosomes for their reuse on the cell surface. The decision depends on ubiquitin moieties that are attached to the Itgb1-tail upon internalization and are either retained and recognized by the ESCRT machinery or removed by DUBs. USP9X, a DUB best known for its role in mitosis and DNA repair (Meng et al, 2023), was shown to remove ubiquitin from α5-integrin tails and stabilize α5β1 integrins in starved cells treated with soluble FN (Kharitidi et al, 2015). Since we excluded an involvement of USP9X in controling α5β1 integrin in cells cultured under steady-state conditions, we decided to perform a Crispr/Cas9-based genetic screen and a BioID-based proximitome screen to identify DUB(s) that maintain the steady-state surface levels of Itgb1.

Our screens identified a ternary deubiquitinase (DUB) complex consisting of USP12 (or its paralog USP46) and the accessory proteins WDR48 and WDR20. This complex maintains Itgb1 protein levels in HAP1 cells, MDA-MB-231 cells, and mouse fibroblasts. In search for a mechanistic explanation for the Itgb1 protein stabilization, we hypothesized that the activity of the USP12/46-WDRs complex removes ubiquitin from Itgb1 at early endosomal membranes and, thereby, inhibits ESCRT-mediated sorting of Itgb1 into intraluminal vesicles (ILVs) and lysosomal degradation, and instead retrieves Itgb1 into recycling endosomes. Consistently, the loss of the two DUBs, USP12, and USP46, which compensate each other, or of either WDR20 or WDR48, which facilitates the DUB activity, decreases the protein stability of Itgb1. Importantly, our cell surface proteome analysis also revealed that the USP12/46-WDRs complex regulates not only the stabilization of Itgb1 but also of other surface proteins, including signaling receptors and solute transporters. This diverse group of surface proteins possessed different cytoplasmic domains and

shared neither sequences/motifs with the Itgb1 tail nor with known nuclear and cytosolic substrates of USP12/46-WDRs (Niu et al, 2023). Immunostaining and proteomics analysis indicate that the USP12/46-WDRs complex-mediated Itgb1 deubiquitination occurs at endosomes, as neither our immunostainings nor previous studies on the adhesome composition found the complex in integrin adhesion sites.

The genetic as well as poximitome screenings neither identified USP9X nor USP10, which have been reported to deubiquitinate α5β1 integrins (Gillespie et al, 2017; Kharitidi et al, 2015). In the case of USP9X, it was shown that a brief treatment of cells with FN was shown to induce USP9X-mediated α5β1 integrin deubiquitination and recycling back to the cell surface (Kharitidi et al, 2015), whereas in our screenings the continuous exposure of the cells to FN-rich serum had no apparent consequences for α5β1 integrin stability. Similarly, the depletion of USP10 decreased Itgb1 protein abundance in corneal stromal fibroblasts (Gillespie et al, 2017), which was not observed in our experimental setup.

Our findings also show that the three lysine residues, K774, K784, and K794, in the Itgb1-tail are ubiquitin-modified in the absence of USP12/46. The K784 is located adjacent to the membrane-proximal NPIY motif that serves as a binding site for Talins, and the K794 is located in the distal NPKY motif that serves as binding site for Kindlin and SNX17. Although we did not succeed to synthesize site-specific ubiquitinated integrin tail peptides to test Kindlin and SNX17 binding, it is conceivable that the ubiquitination of K784 and K794 couples integrin inactivation by compromising Talin and Kindlin binding and blockade of endosomal retrieval and recycling by compromising SNX17 binding with ESCRT-mediated degradation. Hence, Itgb1-tail ubiquitination may have a dual function by coordinating the activity and stability of integrins.

USP12 and USP46 can functionally compensate for each other in vitro. The two DUBs are ubiquitously expressed, however, they

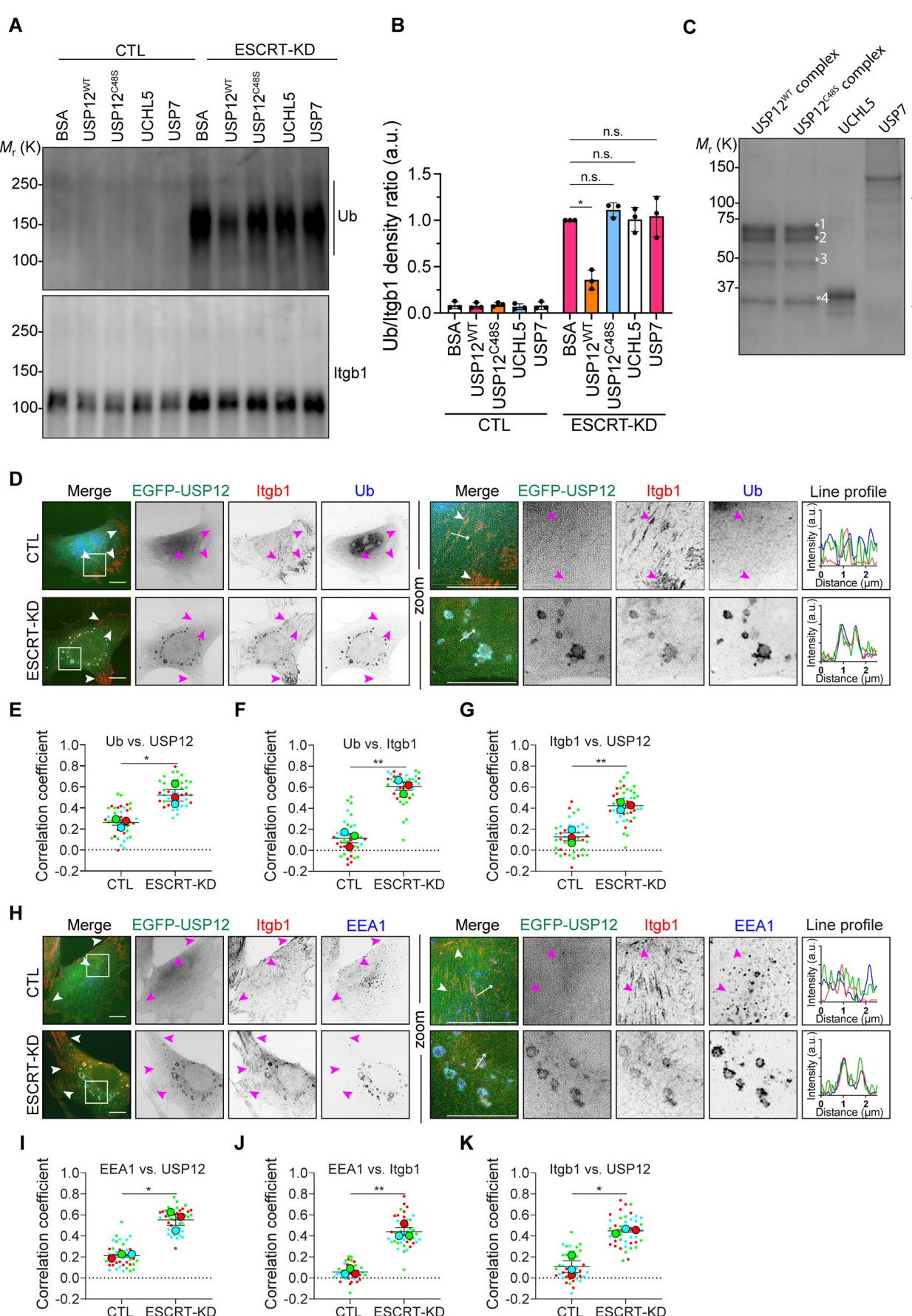

**Figure 6.  The USP12/46-WDRs complex deubiquitinates Itgb1 on endosomes.**

(A, B) In vitro deubiquitination assay using recombinant WDR48-WDR20-USP12 (WT or C48S) complex, UCHL5 or USP7, and ubiquitinated Itgb1 enriched from USP12/46-dKO[ESCRT-KD] fibroblast cell lysate followed by WB (A) for ubiquitin with quantification (B) of Itgb1 ubiquitination levels. The intensity of the Ub signals was normalized to the intensity of the IP-ed Itgb1 signals. Statistical analysis was carried out by RM two-way ANOVA with Dunnett's multiple comparison test comparing the BSA group with USP12[WT], USP12[C48S], UCHL5, or USP7 group in ESCRT-KD condition ($P = 0.0194$, 0.2707, 0.9997, and 0.9844, respectively). *$P < 0.05$; n.s. not significant. Data were shown as Mean ± SD, $n = 3$ independent experiments. BSA served as a negative control. (C) Coomassie-blue staining of recombinant proteins used in the in vitro deubiquitination assay. MS was used to determine the identity of the protein bands indicated in the DUB complexes. Band *1, WDR48; band *2, WDR20; band *3, a cleaved WDR20; band *4, USP12. (D) Representative structured illumination microscopy (SIM) images of Itgb1 and Ub in USP12/46-dKO fibroblasts stably expressing EGFP-USP12 treated with or without ESCRT-KD siRNAs. Boxes indicate magnified cell regions displayed in the Zoom panel. White/pink arrowheads show the Itgb1-labeled FAs, and white arrows show the direction of line profiles. Scale bar, 10 μm. (E-G) Superplots showing the PCC between Ub and EGFP-USP12 (E), Ub and Itgb1 (F), and Itgb1 and EGFP-USP12 (G) in USP12/46-dKO fibroblasts stably expressing EGFP-USP12 and treated with or without ESCRT-KD siRNAs. Statistical analysis was carried out by two-sided Welch's *t*-test. The *P* values in (E-G) are 0.0287, 0.0011, and 0.0048, respectively. *$P < 0.05$; **$P < 0.01$. Data were shown as Mean ± SD, $n = 3$ independent experiments, in total 42 cells per condition. (H) Representative SIM images of Itgb1 and EEA1 in USP12/46-dKO fibroblasts stably expressing EGFP-USP12 treated with or without ESCRT-KD siRNAs. Boxes indicate magnified cell regions displayed in the Zoom panel. White/pink arrowheads show the Itgb1-labeled FAs, and white arrows show the direction of line profiles. Scale bar, 10 μm. (I-K) Superplots showing the PCC between EEA1 and EGFP-USP12 (I), EEA1 and Itgb1 (J), and Itgb1 and EGFP-USP12 (K) in USP12/46-dKO fibroblasts stably expressing EGFP-USP12 and treated with or without ESCRT-KD siRNAs. Statistical analysis was carried out by two-sided Welch's *t*-test. The *P* values in (I-K) are 0.0178, 0.0040, and 0.0208, respectively. *$P < 0.05$; **$P < 0.01$. Data were shown as Mean ± SD, $n = 3$ independent experiments, in total 42 cells per condition. Source data are available online for this figure.

display distinct expression levels in different tissues. USP12 is prominently expressed in bone marrow, whereas USP46 in muscle and brain tissues (proteinatlas.org) (Uhlen et al, 2015). Despite the different abundance of USP46 and USP12 in tissues, USP12- as well as USP46-KO mice are viable and lack obvious developmental and postnatal defects (mousephenotype.org) (Groza et al, 2023). Interestingly, however, increased levels of USP12 and USP46 have been associated with the progressions of several cancers, including breast cancer, liver cancer, and multiple myeloma (Niu et al, 2023). The diminished Itgb1 levels in USP12/46-dKO MDA-MB-231 cells severely impaired migration and invasion. Hence, the link between USP12/46 and Itgb1 stability may well have a contributory role in the course of different cancer entities, originating of both epithelial and blood origin, and call for the exploration of compounds that inhibit the activity of the DUB complex.

# Methods

### Reagents and tools table

| Reagent/resource | Reference or source | Identifier or catalog number |
|---|---|---|
| **Experimental models** | | |
| HAP1 cells | Horizon Discovery | C859 |
| RPE-1 cells | ATCC | CRL-4000 |
| Hela cells | ATCC | CCL-2 |
| MDA-MB-231 cells | ATCC | HTB-26 |
| Itgb1-floxed mouse kidney fibroblasts | Böttcher et al, 2012 | Prof. Dr. Reinhard Fässler, MPIB, Germany |
| Itgb1-null mouse kidney fibroblasts | Böttcher et al, 2012 | Prof. Dr. Reinhard Fässler, MPIB, Germany |
| **Recombinant DNA** | | |
| lentiCas9-blast | Sanjana et al, 2014 | N/A |
| pSpCas9(BB)-2A-Puro (PX459) V2.0 | Addgene | #62988 |

| Reagent/resource | Reference or source | Identifier or catalog number |
|---|---|---|
| PRetroQ-N1-Itga5-miniTurbo | This study | N/A |
| HSC1-human Itgb1-IRES-human Itga5 | This study | N/A |
| pRetroQ-C1-USP12-WT | This study | N/A |
| pRetroQ-C1-USP12[C48S] | This study | N/A |
| pRetroQ-C1-USP12[E190K] | This study | N/A |
| pRetroQ-C1-USP12[V279A/F287A] | This study | N/A |
| pRetroQ-C1-USP12[E190K/V279A/F287A] | This study | N/A |
| pRetroQ-C1-USP46-WT | This study | N/A |
| pRetroQ-C1-USP46[C44S] | This study | N/A |
| pRetroQ-C1-WDR48-WT | This study | N/A |
| pRetroQ-C1-WDR48[K214E/W256A/R272D] | This study | N/A |
| pRetroQ-N1-WDR20-WT | This study | N/A |
| pRetroQ-N1-WDR20[F262A/W306A] | This study | N/A |
| pEGFP-N1-Ub-WT | This study | N/A |
| pEGFP-N1-Ub-K48R | This study | N/A |
| pEGFP-N1-Ub-K63R | This study | N/A |
| pCoofy expression vectors | Addgene | #43974 |
| **Antibodies** | | |
| Rabbit anti-α5 integrin | Cell Signaling Technology | #4705 |
| Mouse anti-human β1 integrin | BD biosciences | Clone 18 |
| Rabbit anti-mouse β1 integrin | Azimifar et al, 2012 | Prof. Dr. Reinhard Fässler, MPIB, Germany |

| Reagent/resource | Reference or source | Identifier or catalog number |
|---|---|---|
| Mouse anti-GAPDH | Calbiochem | CB1001 |
| Mouse anti-Flag | Sigma | clone M2 |
| Chicken anti-GFP | Invitrogen | A10262 |
| Rabbit anti-SNX17 | Proteintech | 10275-1-AP |
| Rat anti-haemagglutin (HA) | Roche | clone 3F10 |
| Mouse anti-WDR48 | Santa Cruz Biotechnology | sc-514473 |
| Mouse anti-WDR20 | Santa Cruz Biotechnology | sc-100900 |
| Rabbit anti-HGS | Proteintech | 10390-1-AP |
| Mouse anti-TSG101 | Santa Cruz Biotechnology | sc-7964 |
| Mouse anti-ubiquitin | Cell Signaling Technology | clone P4D1 |
| Rabbit anti-RFP | MBL Life Science | PM005 |
| Mouse anti-human β₁ integrin | Biolegend | #303004 |
| Hamster anti-mouse β₁ integrin | Biolegend | HMbeta1-1 |
| Rat anti-mouse α₅ integrin | BD Pharmingen | 5H10-27 |
| Hamster anti-human β₁ integrin | BD Pharmingen | Ha2/5 |
| Mous anti-human α₅ integrin | BD Pharmingen | IIA1 |
| Mous anti-human α₅ integrin | BD biosciences | VC5 |
| Rat anti-mouse Lamp1 | DSHB | 1D4B |
| Mouse anti-ubiquitin | Enzo Life Sciences | UBCJ2 |
| Mouse anti-EEA1 | Proteintech | 1E11E4 |
| Donkey anti-rat Alexa 488 | Invitrogen | A21208 |
| Goat anti-rabbit Alexa 568 | Invitrogen | A11036 |
| Donkey anti-mouse Alexa 568 | Invitrogen | A10037 |
| Donkey anti-mouse Alexa 647 | Invitrogen | A31571 |
| **Oligonucleotides and other sequence-based reagents** | | |
| Real-time PCR primers | This study | N/A |
| sgRNAs and genomic PCR primers | This study | Table 1 |
| siRNAs targeting human UPS9X | Dharmacon | L-006099-00-0005 |
| siRNAs targeting mouse UPS9X | Dharmacon | L-046869-01-0005 |
| siRNAs targeting HGS | Dharmacon | L-055516-01-0005 |
| siRNAs targeting TSG101 | Dharmacon | L-049922-01-0005 |
| non-targeting siRNAs | Dharmacon | D-001810-10-05 |
| **Chemicals, Enzymes and other reagents** | | |
| TRITC-conjugated phalloidin | Sigma | P1591 |

| Reagent/resource | Reference or source | Identifier or catalog number |
|---|---|---|
| Hoechst 33342 | Thermo Fischer | #H1399 |
| N-ethylmaleimide | Sigma | E-3876 |
| Protease inhibitor cocktail | Roche | 4693159001 |
| DMSO | Thermo Fisher | 042780.AK |
| MG132 | Sigma | 474787 |
| Bafilomycin A1 | Adipogen | BVT-0252 |
| Cycloheximide | Santa Cruz Biotechnology | sc-3508A |
| Mytomycin C | Sigma | M4287 |
| Poly-L-lysine solution | Sigma | P4707 |
| IMDM medium | Gibco | #31980030 |
| FBS | Gibco | #A5256701 |
| DMEM medium | Gibco | #61965059 |
| BD CytofixTM Fixation Buffer | BD | #554655 |
| AIM-V medium | Gibco | #12055 |
| RPMI-1640 medium | Gibco | #61870 |
| NEAA | Gibco | #11140 |
| DMEM medium | Carl Roth | Nr.9007.1 |
| EZ-Link™ Sulfo-NHS-SS-Biotin | Thermo Fisher | 21217 |
| Streptavidin Mag Sepharose beads | Cytiva | #28985799 |
| TRIzol™ Reagent | Invitrogen | 15596026 |
| iScript™ cDNA Synthesis Kit | Bio-Rad | 1708896 |
| iQ SYBR Green Supermix | Bio-Rad | #1708880 |
| BCA assay kit | Thermo Fisher | #23225 |
| PVDF membranes | Millipore | IPVH00010 |
| Immobilon Western Chemiluminescent HRP substrate | Millipore | #WBKLS0500 |
| EZ-Link™ Sulfo-NHS-LC-Biotin | Thermo Fischer | #21335 |
| EZ-Link™ Sulfo-NHS-SS-Biotin | Thermo Fisher | #21331 |
| MesNa | Sigma | #63705 |
| Iodoacetamide | Sigma | #I6125 |
| GFP nano-trap beads | Chromotek | gta |
| DSP | Thermo Fischer | #22585 |
| Protein A/G agarose beads | Santa Cruz Biotechnology | sc-2003 |
| Recombinant UCHL5 | Novus Biologicals | NBP1-72315 |
| Recombinant USP7 | Novus Biologicals | E-519-025 |
| 2-well silicone insert | ibidi | 81176 |
| Transwell chamber | Corning | 353097 |
| Matrigel invasion chamber | Corning | 354483 |

| Reagent/resource | Reference or source | Identifier or catalog number |
|---|---|---|
| **Software** | | |
| R | https://cran.r-project.org/bin/windows/base/ | |
| Graphpad Prism v10 | https://www.graphpad.com/ | |
| Fiji Image J | https://imagej.net/software/fiji/downloads | |
| Image Lab version 6.1 | https://www.bio-rad.com/ | |
| FlowJo software version 10.10. | https://www.flowjo.com/solutions/flowjo/downloads | |
| Zen (black version) | https://www.zeiss.com/microscopy/en/products/software/zeiss-zen.html | |
| **Other** | | |
| Illumina NextSeq 500 | Illumina | |
| Zeiss LSM780 | Zeiss | |
| Zeiss Elyra PS.1 | Zeiss | |
| EVOS FL Auto2 microscope | Thermo Fisher | |

## Cell culture

HAP1 cells were cultured in an IMDM medium supplemented with 10% FBS. RPE-1 cells, Hela cells, MDA-MB-231 cells, and mouse kidney fibroblasts (Böttcher et al, 2012) in DMEM supplemented with 10% FBS. All cells were cultured at 37 °C with 5% $CO_2$ and routinely tested for mycoplasma.

## CRISPR screen

The CRISPR screen for human DUBs was carried out as previously described (Paulmann et al, 2022) with slight modifications. Briefly, $3 \times 10^6$ HAP1 cells lentivirally transduced (lentiCas9-blast) to stably express the Cas9 protein (Sanjana et al, 2014) and cultured in the presence of 8 ug/ml protamine sulfate were retrovirally transduced with pooled sgRNA library targeting 98 DUBs and five genes essential for cell survival with three sgRNAs per gene and contained 12 non-targeting sgRNAs (Paulmann et al, 2022). Mutagenized HAP1 cells were cultured for one week, sorted for GFP expression by flow cytometry, and expanded for another week. Subsequently, $40 \times 10^6$ cells were harvested, resuspended in 0.4 ml FACS buffer (PBS containing 2% FBS and 2.5 mM EDTA), stained with PE-labeled anti-Itgb1antibody (#303004, Biolegend) for 45 min on ice, washed twice with ice-cold PBS, fixed with BD Cytofix™ Fixation Buffer for 10 min on ice and then for 10 min at room temperature (RT). The fixed cells were stored in FACS buffer supplemented with 0.01% sodium azide at 4 °C in the dark before cells were sorted with a FACSAriaIII flow cell sorter. Prior to loading onto the sorter, cells were stained with Hoechst 33343 for 30 min on ice and haploid cells were gated based on the Hoechst signal followed by forward and sideward gating, and finally gating 5% high and 5% low PE-positive Itgb1 expressing cell populations.

Genomic DNA from the 5% high and 5% low populations was extracted followed by the amplification of the sgRNA cassettes by a two-step PCR approach and by adding adapters and sample barcodes for deep sequencing. PCR products were sequenced on an Illumina NextSeq 500 instrument and the fastq output files were analyzed on the Galaxy platform (Galaxy, 2022) using the instances at usegalaxy.eu. Reads were mapped into the sgRNA library and the count table were analyzed with MAGeCK packages. Figure was generated using the R software with the packages tidyverse, ggplot, and ggrepel.

## Generation of cell lines

The Crispr-Cas9 technique was used to generate knockout (KO) clones of USP12, USP46, WDR48, WDR20, and SNX17 following published protocols (Ran et al, 2013). The vector pSpCas9(BB)-2A-Puro (PX459) V2.0 (a gift from Feng Zhang) was used to express the gRNA and Cas9. The gRNA sequences and primers for genomic DNA amplification and sequencing are listed in Table 1.

The gene disruptions were confirmed by sequencing the genomic DNA regions which were targeted by the gRNAs or Western blot if specific antibodies were available. Itgb1-KO fibroblasts were generated by adding adenoviral Cre to Itgb1-floxed fibroblasts derived from the kidneys of adult Itgb1-floxed mice (Böttcher et al, 2012).

To express ectopic integrin, Itgb1-null fibroblasts were retrovirally transduced with cDNAs encoding human Itgb1 and Itga5. WT or nullizygous USP12, USP46, and WDR48 cells were reconstituted by retrovirally transducing the cells with cDNA constructs tagged with EGFP, mScarlet or 3XFlag tags, respectively, as indicated in the Results section. To generate cells stably expressing miniTurbo-tagged Itga5, Itgb1-floxed fibroblasts were retrovirally transduced with the human Itga5 tagged with the miniTurbo DNA at the 3'-end.

## Plasmids and siRNAs

The mouse USP12, USP46, and WDR48 cDNAs were reverse transcribed from mouse kidney fibroblasts and cloned into pEGFP-C1 and pRetroQ-C1 vectors (Clontech). The mouse WDR20 cDNA was prepared as described above and cloned into pEGFP-N1 (Clontech). The cDNAs encoding catalytically inactive USP12[C48S] and USP46[C44S] and binding deficient USP12[E190K], USP12[V279A/F287A], USP12[E190K/V279A/F287A], WDR48[K214E/W256A/R272D], and WDR20[F262A/W306A] were generated by site-directed mutagenesis and cloned into pEGFP-C1 and pRetroQ-C1 vectors (Clontech). The cDNAs encoding WDR48[1-580] and WDR48[1-359] deletion mutants were amplified from the WDR48[WT] cDNA and cloned into the pRetroQ-C1 vector. The Itgb1 and Itga5 cDNAs carrying lysine for arginine substitutions in the cytoplasmic domain (Itgb1[8xKR] and Itga5[4xKR]) were previously generated by site-directed mutagenesis (Böttcher et al, 2012) and cloned into the HSC1 retroviral vector (a gift from James Ellis, Addgene plasmid # 58254). The cDNA encoding miniTurbo-tagged Itga5 was generated by fusing the miniTurbo tag (a gift from Alice Ting, Addgene plasmid # 107168) in frame to the 3′ end of the human Itag5 cDNA (a gift from Rick Horwitz, Addgene plasmid # 15238) and subsequently cloned into pRetroQ-N1 vector. The cDNAs encoding Ub-WT (a gift from Ted Dawson, Addgene plasmid # 17608), Ub-K48R (a gift from Ted Dawson, Addgene plasmid # 17604), and Ub-K63R (a gift from

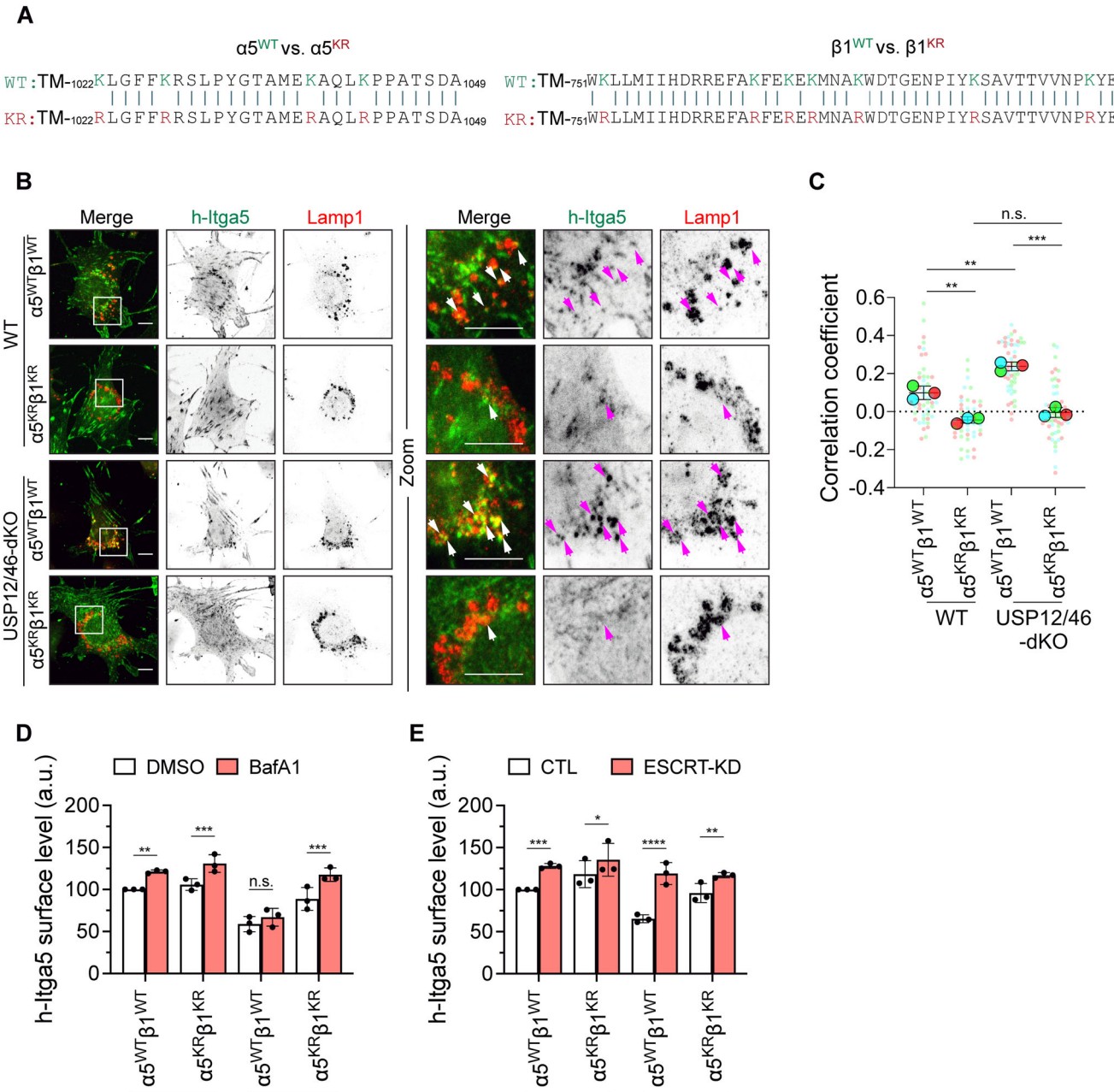

**A**

α5^WT vs. α5^KR

WT:TM-_{1022}KLGFFKRSLPYGTAMEKAQLKPPATSDA_{1049}

KR:TM-_{1022}RLGFFRRSLPYGTAMERAQLRPPATSDA_{1049}

β1^WT vs. β1^KR

WT:TM-_{751}WKLLMIIHDRREFAKFEKEKMNAKWDTGENPIYKSAVTTVVNPKYEGK_{798}

KR:TM-_{751}WRLLMIIHDRREFARFERERMNARWDTGENPIYRSAVTTVVNPRYEGR_{798}

Cecile Pickart, Addgene plasmid # 18898) were cloned into pEGFP-N1 vector (Clontech).

Depletion of USP9X was carried out with siRNAs pools targeting human and mouse USP9X in mouse fibroblasts, RPE-1, Hela, and MDA-MB-231 cells. The transduced cells were detached, 48 h later, washed twice with PBS, and seeded on PLL-coated 12-well plates in DMEM supplemented with 10% FBS or in FN-coated 12-well plates in serum-replacement medium (Benito-Jardon et al, 2017) containing 46.5% AIM-V medium, 5% RPMI-1640 medium, 1% non-essential amino acid (NEAA), and 47.5% DMEM medium and cultured overnight at 37 °C. The surface levels of Itgb1 were assessed by flow cytometry and the total levels by Western blot.

To disrupt the ESCRT function, a co-depletion was performed using siRNAs targeting HGS and TSG101 (Lobert et al, 2010). A non-targeting siRNA was used as a control.

## Transient transfection and viral transduction

Cells were transfected with plasmids using Lipofectamine 2000 or with siRNA using Lipofectamine RNAiMAX following the manufacturer's protocol (Invitrogen). Viral transduction of expression constructs to generate stable cell lines was carried out as described previously (Theodosiou et al, 2016).

**Figure 7. Itgb1 ubiquitination is required for Itgb1 sorting into the endosomal lumen.**

(A) Amino acid sequence alignments of the cytosolic tails of WT α5β1 integrin (α5$^{WT}$β1$^{WT}$) and mutant α5$^{KR}$β1$^{KR}$ where lysine residues were substituted for non-ubiquitinated arginine residues. (B) Representative IF images of exogenous human Itga5 (h-Itga5) and Lamp1 in BafA1-treated Itgb1-KO$^{WT}$ and Itgb1-KO$^{USP12/46-dKO}$ fibroblasts. Itgb1-KO cells were retrovirally transduced with human α5$^{WT}$β1$^{WT}$ or α5$^{KR}$β1$^{KR}$ integrins. A human-specific Itga5 antibody was used for IF. Boxes indicate magnified cell regions displayed in the Zoom panel. White/pink arrowheads indicate the colocalization of h-Itga5 with Lamp1. Sum intensity projections from confocal stacks are shown. Scale bar, 10 µm. (C) Superplots showing the PCC between Itgb1 and Lamp1 in fibroblasts as described above. Statistical analysis was carried out by ordinary two-way ANOVA with Šidák's multiple comparison test comparing the α5$^{WT}$β1$^{WT}$ group with α5$^{KR}$β1$^{KR}$ group in Itgb1-KO$^{WT}$ fibroblasts ($P = 0.0028$) and in Itgb1-KO$^{USP12/46-dKO}$ fibroblasts ($P = 0.0001$); comparing the α5$^{WT}$β1$^{WT}$ group between Itgb1-KO$^{WT}$ and Itgb1-KO$^{USP12/46-dKO}$ fibroblasts ($P = 0.0033$); the α5$^{KR}$β1$^{KR}$ group between Itgb1-KO$^{WT}$ and Itgb1-KO$^{USP12/46-dKO}$ fibroblasts ($P = 0.4314$). **$P < 0.01$; ***$P < 0.001$; n.s. not significant. Data were shown as Mean ± SD, $n = 3$ independent experiments; 48 cells were analyzed in α5$^{WT}$β1$^{WT}$-expressing Itgb1-KO$^{WT}$ cells, 44 in α5$^{KR}$β1$^{KR}$-expressing Itgb1-KO$^{WT}$ cells, 52 in α5$^{WT}$β1$^{WT}$-expressing Itgb1-KO$^{USP12/46-dKO}$ cells and 55 in α5$^{KR}$β1$^{KR}$-expressing Itgb1-KO$^{USP12/46-dKO}$ cells. (D) Human Itga5 surface levels in DMSO- or BafA1-treated cells as described above. Statistical analysis was carried out by RM two-way ANOVA with Šidák's multiple comparison test comparing the DMSO group with BafA1 group in α5$^{WT}$β1$^{WT}$-expressing Itgb1-KO$^{WT}$ fibroblasts ($P = 0.0017$); in α5$^{KR}$β1$^{KR}$-expressing Itgb1-KO$^{WT}$ fibroblasts ($P = 0.0006$); in α5$^{WT}$β1$^{WT}$-expressing Itgb1-KO$^{USP12/46-dKO}$ fibroblasts ($P = 0.2153$); and in α5$^{KR}$β1$^{KR}$-expressing Itgb1-KO$^{USP12/46-dKO}$ fibroblasts ($P = 0.0002$). **$P < 0.01$; ***$P < 0.001$; n.s. not significant. Data were shown as Mean ± SD, $n = 3$ independent experiments. (E) Human Itga5 surface levels in cells, as described above, treated with or without ESCRT-KD siRNAs. Statistical analysis was carried out by RM two-way ANOVA with Šidák's multiple comparison test comparing the CTL group with ESCRT-KD group in α5$^{WT}$β1$^{WT}$-expressing Itgb1-KO$^{WT}$ fibroblasts ($P = 0.0017$); in α5$^{KR}$β1$^{KR}$-expressing Itgb1-KO$^{WT}$ fibroblasts ($P = 0.0006$); in α5$^{WT}$β1$^{WT}$-expressing Itgb1-KO$^{USP12/46-dKO}$ fibroblasts ($P < 0.0001$); and in α5$^{KR}$β1$^{KR}$-expressing Itgb1-KO$^{USP12/46-dKO}$ fibroblasts ($P = 0.0042$). *$P < 0.05$; **$P < 0.01$; ***$P < 0.001$; ****$P < 0.0001$; n.s. not significant. Data were shown as Mean ± SD, $n = 3$ independent experiments. Source data are available online for this figure.

## Proximity-dependent biotin identification

To perform proximity-dependent biotin identification, Itgb1 floxed and Itgb1-KO cells stably expressing miniTurbo-tagged Itga5 were cultured in three 15-cm dishes, grown to 80% confluence, incubated with 50 µM biotin in PBS solution for 30 min at 37 °C, washed twice with PBS and then incubated with DMEM for a further 10 min. After cells were washed twice with PBS, lysates were generated with lysis buffer (0.1% SDS, 0.1% SDC, 1% Triton, 150 mM NaCl, 50 mM Tris, pH = 8) containing protease inhibitor cocktail, then incubated with streptavidin magnetic beads (Cytiva) for 2.5 h at 4 °C. The beads were washed three times with lysis buffer, once with PBS to remove detergent, and then incubated in SDC buffer comprising of 1% sodium deoxycholate (SDC, Sigma), 40 mM 2-chloroacetamide (CAA, Sigma), 10 mM tris (2-carboxyethyl) phosphine (TCEP; Thermo Fisher), and 100 mM Tris, pH 8.0 at 37 °C. After a 20-min incubation at 37 °C, the samples were diluted at a 1:2 ratio with MS grade water (VWR), the proteins digested overnight at 37 °C with 0.5 µg of trypsin (Promega) and the resulting supernatant containing the peptide mixture harvested using a magnetic rack, acidified with trifluoroacetic acid (TFA; Merck) to achieve a final concentration of 1% and desalted using SCX StageTips. The samples were eluted, vacuum-dried, and reconstituted in LC-MS grade water containing 0.1% formic acid before being loaded onto Evotips (Evotip Pure, Evosep).

An LC-MS/MS system coupled to a timsTOF Pro mass spectrometer (Bruker) was used to analyze the peptides. Evotips eluates were applied onto a 15-cm column (PepSep C18 15 cm × 15 cm, 1.5 µm, Bruker Daltonics) utilizing the Evosep One HPLC system. The column temperature was set to 50 °C, peptide separation was achieved with the 30 SPD (samples per day) method, eluted and directly introduced into the timsTOF Pro mass spectrometer (Bruker Daltonics) via the nanoelectrospray interface. Data acquisition on the timsTOF Pro instrument was performed via the timsControl software. The MS functioned in data-dependent PASEF mode, performing one survey TIMS-MS scan and ten PASEF MS/MS scans per acquisition cycle. The analysis spanned a mass scan range of 100–1700 m/z and an ion mobility range from $1/K0 = 0.85$ to $1.35$ Vs cm$^{-2}$, with uniform ion accumulation and ramp time of 100 ms each in the dual TIMS analyzer, achieving a spectra rate of 9.43 Hz. Precursor ions suitable for MS/MS analysis were isolated within a 2 Th window for m/z < 700 and 3 Th for m/z > 700 by promptly adjusting the quadrupole position as precursors eluted from the TIMS device. The collision energy was adapted based on ion mobility, commencing at 45 eV for $1/K0 = 1.3$ Vs cm$^{-2}$ and decreasing to 27 eV for 0.85 Vs cm$^{-2}$. Collision energies were interpolated linearly between these thresholds and remained constant above or below them. Uniquely charged precursor ions were filtered out employing a polygon filter mask, and supplementary m/z and ion mobility details were harnessed for "dynamic exclusion" to avert the re-evaluation of precursors once they attained a "target value" of 14500 a.u.

The MaxQuant computational platform (version 2.2.0.0) (Cox and Mann, 2008) was used to analyze the raw data with typical configurations tailored for Orbitrap or ion mobility data. Essentially, the peak list was cross-referenced against the Uniprot database of mus musculus (downloaded in 2023). Cysteine carbamidomethylation was designated as a fixed modification, while methionine oxidation and N-terminal acetylation were considered variable modifications. Protein quantification across runs was performed utilizing the MaxLFQ algorithm.

## Flow cytometry

To assess surface integrin levels, cells were trypsinized, washed with cold PBS, incubated with FACS antibodies diluted 1:400 in PBS supplemented with 1% BSA for 30 min on ice in the dark, washed with cold PBS, and analyzed using the BD LSRFortessa™ X-20 Cell Analyzer. The geometric mean fluorescence intensity of each sample was then evaluated using the FlowJo software version 10.10.

## Cell surface proteome

Cells were cultured in three independent 6-cm dishes at ~80% confluence. Cells were surface biotinylated with 0.2 mg/ml EZ-Link™ Sulfo-NHS-SS-Biotin in cold PBS for 30 min at 4 °C. Cell lysates were generated by incubating cells in lysis buffer (0.1% SDS, 0.1% SDC, 1% Triton, 150 mM NaCl, 50 mM Tris, pH = 8) with protease inhibitor cocktail. The cell lysate was then incubated

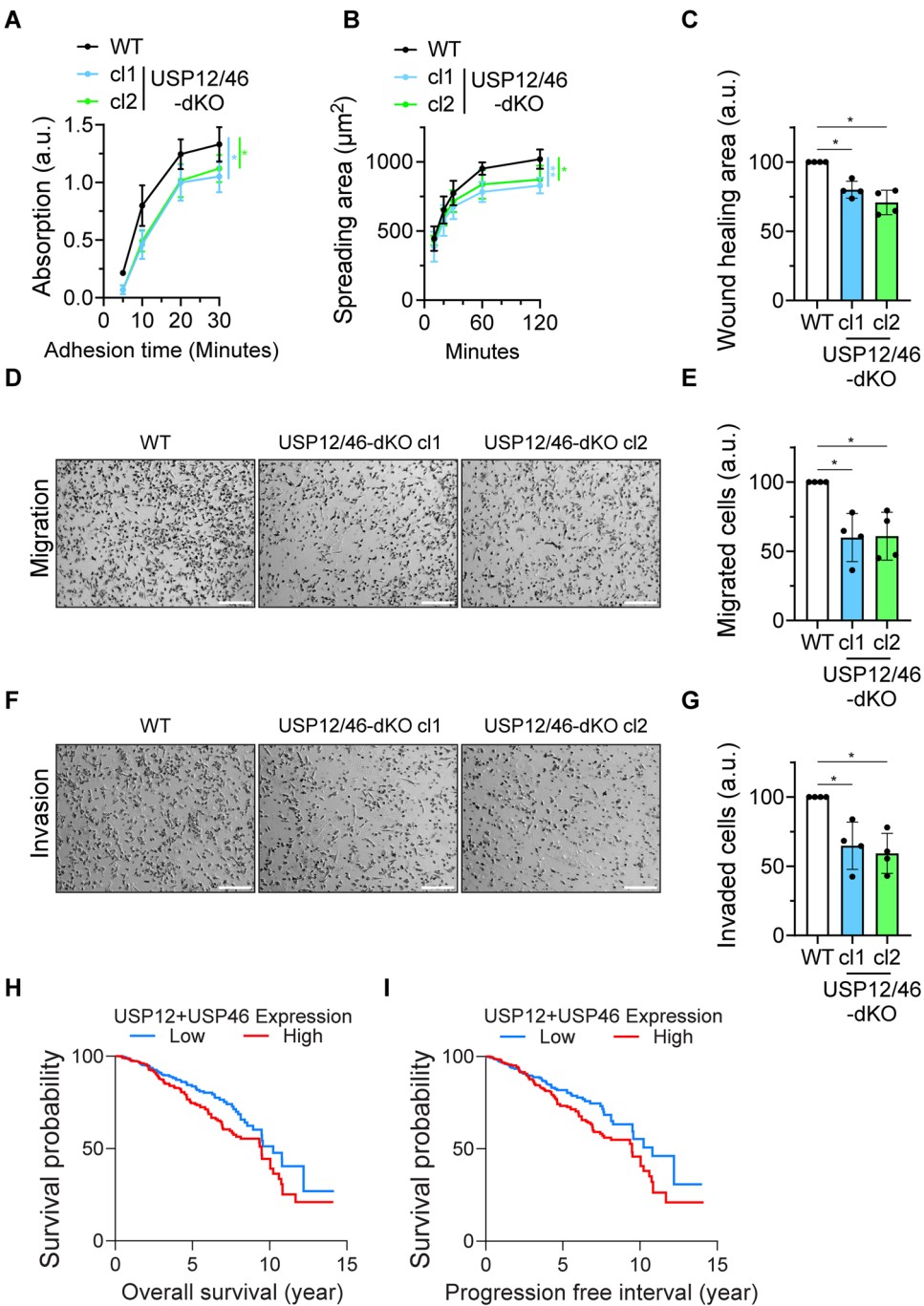

with Streptavidin Mag Sepharose beads for 2.5 h at 4 °C. The beads were washed three times with lysis buffer and once with PBS to remove detergent. Proteins on beads were digested by trypsin, and the peptides were prepared as described above for proximity-dependent biotin identification. MDA-MB-231 samples were analyzed on the Brucker TimsTOF Pro with procedures and parameters as described above. Mouse fibroblast samples were analysed on a Thermo Fisher QExactive HF mass spectrometer. Peptides were chromatographically separated using a 30-cm analytical column (inner diameter: 75 microns), packed in-house with ReproSil-Pur C18-AQ 1.9-micron beads from Dr. Maisch

GmbH. This separation was achieved through a 60-min gradient ranging from 8 to 30% buffer composed of 80% acetonitrile and 0.1% formic acid. The mass spectrometer operated in a data-dependent mode, with survey scans spanning from 300 to 1650 m/z at a resolution of 60,000 (at m/z = 200). It selectively picked up to 10 of the top precursors for fragmentation using higher energy collisional dissociation (HCD) with a normalized collision energy of 28. MS2 spectra were acquired at a resolution of 30,000 (at m/z = 200). Automatic gain control (AGC) targets for both MS and MS2 scans were set to 3E6 and 1E5, respectively, with maximum injection times of 100 ms for MS scans and 60 ms for MS2 scans.

**Figure 8.   The USP12/46-WDRs complex promotes cancer cell migration and invasion.**

(A) Quantification of cell adhesion of WT and USP12/46-dKO fibroblasts at indicated time points after seeding on FN-coated plates. Statistical analysis was carried out by RM one-way ANOVA with Dunnett's multiple comparison test comparing WT with USP12/46-dKO cl1 or cl2 fibroblasts at the 30-min time point ($P = 0.0103$ and $0.0419$, respectively). *$P < 0.05$. Data were shown as Mean ± SD, $n = 4$ independent experiments. (B) Cell spreading area of WT and USP12/46-dKO fibroblasts on FN-coated glass surface shown at indicated time points after seeding. Statistical analysis was carried out by RM one-way ANOVA with Dunnett's multiple comparison test comparing WT with USP12/46-dKO cl1 or cl2 fibroblasts at the 120-min time point ($P = 0.0031$ and $0.0149$, respectively). *$P < 0.05$; **$P < 0.01$. Data were shown as Mean ± SD, $n = 4$ independent experiments. (C) Normalized wound healing area of WT and USP12/46-dKO fibroblasts on FN-coated plates after 12 h. Statistical analysis was carried out by RM one-way ANOVA with Dunnett's multiple comparison test comparing WT with USP12/46-dKO cl1 or cl2 fibroblasts ($P = 0.0120$ and $0.0116$, respectively). *$P < 0.05$. Data were shown as Mean ± SD, $n = 4$ independent experiments. (D, E) Representative images (D) and quantification (E) of WT and USP12/46-dKO MDA-MB-231 cells upon migration through transwell inserts 16 h after seeding. Scale bar, 200 μm. Statistical analysis was carried out by RM one-way ANOVA with Dunnett's multiple comparison test comparing WT with USP12/46-dKO cl1 or cl2 MDA-MB-231 cells ($P = 0.0314$ and $0.0333$, respectively). *$P < 0.05$. Data were shown as Mean ± SD, $n = 4$ independent experiments. (F, G) Representative images (F) and quantification (G) of WT and USP12/46-dKO MDA-MB-231 cells upon migration through Matrigel-coated transwell inserts 16 h after seeding. Scale bar, 200 μm. Statistical analysis was carried out by RM one-way ANOVA with Dunnett's multiple comparison test comparing WT with USP12/46-dKO cl1 or cl2 MDA-MB-231 cells ($P = 0.0422$ and $0.0183$, respectively). *$P < 0.05$. Data were shown as Mean ± SD, $n = 4$ independent experiments. (H, I) Kaplan–Meier plot of overall survival (H) and progression-free interval (I) of breast cancer patients with high (red line) or low (blue line) combined total *USP12* and *USP46* gene expression levels. The GDC TCGA dataset obtained from the UCSC Xena project (Goldman et al, 2020) was used. Two-group risk model with a cut-off at the median was applied. $P$ values were calculated by log-rank test. The $P$ values in (H) and (I) are $0.0197$ and $0.0388$, respectively. Source data are available online for this figure.

The raw data was analyzed as described above in proximity-dependent biotin identification.

## Quantitative PCR

Total RNA was extracted from fibroblasts using TRIzol™ Reagent following the manufacturer's instructions. cDNA was synthesized using iScript™ cDNA Synthesis Kit, followed by real-time PCR using iQ SYBR Green Supermix on LightCycler® 480 (Roche). Primers for GAPDH: forward 5′-AGGTCGGTGTGAACG-GATTTG-3′, reverse 5′- TGTAGACCATGTAGTTGAGGTCA-3′. Primers for Itgb1: forward 5′- ATGCCAAATCTTGCGGAGAAT-3′, reverse 5′- TTTGCTGCGATTGGTGACATT-3′.

## Western blot

Cells were lysed in RIPA buffer (0.1% SDS, 0.1% SDC, 1% Triton, 150 mM NaCl, 50 mM Tris, pH = 8) containing a protease inhibitor cocktail. Protein concentrations were determined using the BCA assay kit. Lysates were boiled at 95 °C for 5 min in 1x Laemmli buffer, resolved on SDS–PAGE gels, and transferred to 0.45 um PVDF membranes. After transfer, the membranes were briefly washed with PBS containing 0.1% Tween-20 (PBST) for 5 min before incubation with 5% BSA in PBST for 1 h at RT, incubated with primary antibodies overnight at 4 °C, washed in PBST and then incubated with HRP-conjugated secondary antibodies for 1 h at RT. A GE Amersham AI600 imager was used to detect the chemiluminescence generated upon the addition of Immobilon Western Chemiluminescent HRP substrate to the membranes, and Image Lab version 6.1 was used to quantify densitometries.

## Integrin degradation, internalization, and recycling assays

These assays were carried out as previously described (Böttcher et al, 2012) with slight modifications. Briefly, for cycloheximide (CHX) chase, fibroblasts in a six-well plate ($4 \times 10^5$/well) were treated with cycloheximide (5 μg/ml) for 0, 12, and 24 h before cell lysis and determined by WB. The degradation curves for Itgb1 was normalized to 0 h-time point for each cell line. For integrin surface stability, fibroblasts were washed with cold PBS and surface biotinylated with EZ-Link™ Sulfo-NHS-LC-Biotin (dissolved in PBS at 0.2 mg/ml) for 30 min on ice. Followed by washing with PBS twice, fibroblasts were cultured for 0, 12, and 24 h before cell lysis using lysis buffer (75 mM Tris, 200 mM NaCl, 7.5 mM EDTA, and 7.5 mM EGTA, 1.5% Triton X-100, 0.75% Igepal CA-630 and protease inhibitor cocktail). Biotinylated Itgb1 was measured by capture-ELISA.

For the internalization assay, fibroblasts were washed with cold PBS and surface biotinylated using EZ-Link™ Sulfo-NHS-SS-Biotin (dissolved in PBS at 0.2 mg/ml) for 30 min on ice. Followed by washing with PBS twice, fibroblasts were incubated in prewarmed DMEM with 10% FBS at 37 °C for the specified time intervals to allow Itgb1 internalization. After internalization, the biotinylated proteins on the cell surface were reduced using MesNa for 20 min on ice. This was followed by washing and quenching with iodoacetamide for 10 min on ice. After quenching, cells were lysed and biotinylated Itgb1 was measured by capture-ELISA.

For the recycling assay, cell surface-labeled integrins were allowed for internalization for 20 min before the MesNa reduction step described in the internalization assay. Fibroblasts were then washed and incubated in DMEM with 10% FBS for the specified time points to allow Itgb1 recycling. Followed by a second round of cell MesNa reduction and subsequent quenching, cells were lysed and biotinylated Itgb1 was measured by capture-ELISA.

## Immunoprecipitation

The GFP-based immunoprecipitation (GFP IP) was carried out as previously described with slight modifications (Chen et al, 2022). In brief, cells were lysed in lysis buffer (50 mM Tris-HCl, 150 mM NaCl, 1 mM EDTA, 1% Triton, protease inhibitor cocktail), and the supernatant was incubated with GFP nano-trap beads for 2 h at 4 °C. The beads were washed three times with the lysis buffer, then boiled in 2x Laemmli buffer, separated onto SDS–PAGE followed by WB for detecting the indicated proteins.

## Itgb1 in vivo crosslinking co-IP

Fibroblasts grown 15-cm dish to 80% confluence were siRNA-treated for 48 h, washed with PBS twice, incubated on ice, then incubated with cross-linker (DSP, 0.1 mg/ml in PBS) solution or

**Table 1. gRNA targeting sequences and primers for genomic PCR and Sanger sequencing for specified genes.**

| Genes | gRNA target sequence | Forward primer for genomic DNA amplification | Reverse primer for genomic DNA amplification | Sequencing primer |
|---|---|---|---|---|
| human USP12 | TCTTGTGATGAACTTCTTAG | GCAGTTTGGGAATACCTGCTAC | GTGCTGGAATTATGGCACACTA | GCAGTTTGGGAATACCTGCTAC |
| human USP46 | TATTGCGGACATCCTTCAGG | TAACCACCTTCTCCTTTCCAGA | GTTTCACAGTTCAAGCATCGAG | TAACCACCTTCTCCTTTCCAGA |
| mouse Usp12 | TTTACAGGGCGCCAATGCCT | TTTGCTGTAACTTGAGTGTGGC | TTTGCTGTGAGAATTCTGTTGC | TTTGCTGTGAGAATTCTGTTGC |
| mouse Usp46 | GCCGTTCCGGGAGAATGTGT | ATTTTGAGGCTACACAGAACCG | GCAGTTTGGAAACACATGCTAC | ATTTTGAGGCTACACAGAACCG |
| mouse Wdr20 | GACCGCCTCTGCTTCAATGT | CGAGATTAAGAGACCCAATTCACCA | GGCCACTCAAAAGTACAAGTG | CGAGATTAAGACCCAATTCACCA |
| mouse Wdr48 | ACATACCGAGTCCATGATGA | GCACCTCACCTTATTTCCTTTG | AGGGTCTTCTTGACCCCATTAT | AGGGTCTTCTTGACCCCATTAT |
| mouse Snx17 | CTCCATGACATCGTGTCAT | | | |

PBS for 30 min on ice, washed and quenched with quenching solution (50 mM Tris, pH 7.4, 150 mM NaCl, 1 mM $MgCl_2$, 1 mM $CaCl_2$). Cells were lysed in lysis buffer (50 mM Tris-HCl, 150 mM NaCl, 1 mM EDTA, and 1% Triton), and the lysates were sonicated and cleared by centrifugation. The supernatant was incubated with anti-β1 integrin antibody (homemade) and protein A/G agarose beads for 3 h at 4 °C. The beads were washed with lysis buffer three times and one more time with PBS to remove the detergent. Peptides were prepared and processed on the MS as described above for proximity-dependent biotin identification. Raw data were analyzed using the Spectronaut 18.0 in directDIA+ (library-free) mode with the peak list was cross-referenced against the Uniprot database of mus musculus (downloaded in 2023). Cysteine carbamidomethylation was designated as a fixed modification, while methionine oxidation and N-terminal acetylation were considered variable modifications. Protein quantification across samples was achieved via label-free quantification (MaxLFQ) at the MS2 level.

## Ubiquitination assay

To measure endogenous ubiquitinated Itgb1, fibroblasts cultured in a 15-cm dish at ~80% confluence in the presence of 10% FBS were treated with non-targeting siRNA or siRNA simultaneously targeting mouse HGS and TSG101 (ESCRT-KD) for 48 h. Cells were collected by scraping with PBS supplemented with 20 mM N-ethylmaleimide. Cell pellets were then lysed with 100 ul of 1% SDS in PBS and immediately boiled for 10 min to denature the proteins. The lysate was then diluted with 900 ul lysis buffer (1% Triton, 150 mM NaCl, 50 mM Tris, pH = 8, 1 mM EDTA, 20 mM N-ethylmaleimide) supplemented with the protease inhibitor cocktail. The diluted lysate was further sonicated and cleared by centrifugation. The supernatant was then incubated with anti-β1 integrin antibody (homemade) and protein A/G agarose beads for 3 h at 4 °C. The beads were then boiled in 2x Laemmli buffer, eluted and the elute was Western blotted for ubiquitin and Itgb1. Alternatively, cells were lysed in RIPA buffer (0.1% SDS, 0.1% SDC, 1% Triton, 150 mM NaCl, 50 mM Tris, 20 mM N-ethylmaleimide, pH = 8) with a protease inhibitor cocktail. The cell lysate was then incubated with the ubiquitin selector beads (N2510, NanoTag) for 3 h at 4 °C to enrich for ubiquitinated proteins. Then, the beads were boiled in 2x Laemmli buffer and the elute was subjected to SDS–PAGE followed by WB for Itgb1.

To assess the ubiquitination sites on the Itgb1 cytoplasmic tail, ubiquitinated proteins were enriched using cell lysate from USP12/46-dKO[ESCRT-KD] fibroblasts. Peptides on the beads were prepared and analyzed by MS as described above for proximity-dependent biotin identification.

To determine the ubiquitin linkage specificity of Itgb1, fibroblasts were cultured in 15-cm dish to 80% confluence, transfected with constructs expressing HA-tagged Ub[WT], Ub[K48R], or Ub[K63R] in the presence or absence of ESCRT-KD siRNA for 48 h, harvested, followed by Itgb1 immunoprecipitation as described above. The beads were boiled in 2x Laemmli buffer, eluted and the elute was subjected to SDS–PAGE followed by WB for HA and Itgb1.

For the in vitro deubiquitination assay, USP12/46-dKO fibroblasts cultured in a 15-cm dish with 80% confluence were treated with or without ESCRT-KD siRNA for 48 h. Cells were

harvested, Itgb1 was immunoprecipitated using the anti-β1 integrin antibody (homemade) coupled protein A/G agarose beads. The agarose beads were incubated with 100 nM recombinant USP12-WDRs complexes (generated as described below), recombinant UCHL5 or USP7 in a reaction buffer containing 40 mM Tris, 0.5 mM EDTA, 100 mM Nacl, 0.1% BSA, 1 mM TCEP (PH = 7.4) for 30 min at 37 °C with constant shaking at 900 rpm in a thermal mixer. The beads were washed three times with washing buffer (1% Triton, 150 mM NaCl, 50 mM Tris, 1 mM EDTA, pH = 8), boiled in 2x Laemmli buffer, eluted, and the elute was subjected to SDS–PAGE followed by WB for ubiquitin and Itgb1.

### Expression and purification of recombinant proteins

Full-length recombinant WDR20 (1–569aa) N-terminally tagged with His6 was expressed in *E.coli*, and WDR48 (1-580 aa) N-terminally tagged with Strep-tag II together with untagged USP12^WT and USP12^C48S (both 40-370aa) were expressed in insect cells as reported previously (Li et al, 2016) using the pCoofy expression vectors (a gift from Sabine Suppmann) and purified to ~80% purity followed previously published protocols (Aretz et al, 2023). The purified USP12-WDR48 complex was incubated with WDR20 overnight to form the ternary DUB complex that was purified by size-exclusion chromatography. The purity of the recombinant proteins was verified by SDS–PAGE followed by Coomassie staining and MS.

### Immunofluorescence microscopy

Cells were grown overnight on FN-coated (5 μg/ml) coverslips that were kept in a 12-well plate ($1 \times 10^5$ per well), fixed with 4% PFA for 15 min at RT, and permeabilized with 0.1% Triton X-100 for 10 min at RT. Cells were blocked with 5% BSA for 1 h at RT, incubated with primary antibodies overnight at 4 °C, then with secondary antibodies (1:400 dilution) for 1 h at RT and finally mounted in Elvanol No-Fade™ Mounting Medium.

Images were captured by the Zeiss LSM780 confocal laser scanning microscope or by the Zeiss Elyra PS.1 structured illumination microscope using ZEN software (Black version). Pearson correlation coefficient (PCC) analysis was performed using Fiji ImageJ software with the EzColocalization plug-in (Stauffer et al, 2018).

### Adhesion, spreading, and wound healing assays

The adhesion and spreading assays were performed as previously described (Theodosiou et al, 2016) with slight modifications. Briefly, fibroblasts were starved for 4 h in DMEM without FBS. Then the cells were trypsinized and incubated for a further 1 h at 37 °C in DMEM supplemented with 3% BSA. The starved cells were then seeded on 96-well plates (40,000 per well) coated with either Poly-L-lysine (1:10 dilution) or fibronectin (5 μg/ml) or 3% BSA and incubated for 5, 10, 20, and 30 min at 37 °C, vigorously washed with PBS, fixed in 4% PFA, stained with 0.1% Crystal Violet solution and then dissolved in 2% SDS. The OD value of each well was acquired by a plate reader at 595 nm. Adhesion capacity was normalized using the equation: Normalized $OD_{595} = (OD_{FN} - OD_{BSA})/OD_{PLL}$.

For the spreading assay, the starved cells were seeded on FN-coated (5 μg/ml) glass coverslips and incubated at 37 °C for the indicated time. Cells were then fixed in 4% PFA and stained with TRITC-conjugated phalloidin (1:4000, 2 h at RT) and Hoechst 33342 (1:10,000, 10 min at RT). Images were captured by the LSM780 confocal microscope, and at least 45 cells per time point were analyzed for cell spreading area using Fiji ImageJ software.

For the wound healing assay, cells were seeded on an FN-coated (5 μg/ml) 24-well plate containing a two-well silicone insert with a cell-free gap of approximately 500 μm overnight ($1 \times 10^4$ cells per well). Then the insert was removed, cells were briefly washed with PBS to remove cell debris and allowed to migrate for 12 h at 37 °C with 5% $CO_2$ in DMEM supplemented with mytomycin C (5 μg/ml) to inhibit cell proliferation. The wound healing area was imaged using an EVOS FL Auto2 microscope and measured using Fiji ImageJ software.

### Transwell migration and invasion assays

Transwell chambers with 8-μm pore-sized membranes and Matrigel invasion chambers with 8-μm pore-sized membranes were used for migration and invasion assays, respectively. Briefly, $5 \times 10^4$ cells in DMEM without FBS were added to the upper chamber of the inserts, while DMEM supplemented with 10% FBS was added to the lower compartment. Cells were incubated for 16 h. Cells on the upper side of the membrane were gently scraped and washed off with PBS before the inserts were immersed in ice-cold methanol for 20 min at RT to fix the cells on the lower side of the membrane. The cells were then stained with 0.1% Crystal Violet solution for 20 min at RT, washed three times with distilled water, and then imaged using a Leica DM IL LED microscope. Cell numbers were quantified in five random fields using Fiji ImageJ software.

### Statistics

Statistical analyses were performed using GraphPad Prism v.10 (GraphPad Software). Tests used, multi-comparison correction methods, calculated *P* values, and significance cutoff were indicated in the figure legends for each quantification.

## Data availability

The datasets produced in this study are available in the following databases: CRISPR screen data have been deposited in the NCBI Sequence Read Archive under accession number PRJNA1146728. The mass spectrometry proteomics data have been deposited to the ProteomeXchange Consortium via the PRIDE (Perez-Riverol et al, 2022) partner repository with the dataset identifier PXD054760 (http://www.ebi.ac.uk/pride/archive/projects/PXD054760).

The source data of this paper are collected in the following database record: biostudies:S-SCDT-10_1038-S44319-024-00300-9.

## Peer review information

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

## Acknowledgements

We thank the sequencing, mass spectrometry, and imaging facilities of the Max Planck Institute of Biochemistry for their invaluable support, and Dr. Arnoud Sonnenberg for critically reading the manuscript and discussions. This work was supported by the German Research Foundation (DFG)—DFG BA 2851/6-1 (project ID: 452409123) and DFG BA 2851/7-1 (project ID: 537477296) to FB and the European Research Council (ERC) under the European Union's Horizon 2020 research and innovation program (grant agreement No. 810104—Point) and the Max Planck Society to RF.

## Author contributions

**Kaikai Yu**: Investigation; Writing—original draft; Writing—review and editing. **Guan M Wang**: Conceptualization; Supervision; Investigation; Writing—original draft. **Shiny Shengzhen Guo**: Investigation. **Florian Bassermann**: Resources. **Reinhard Fässler**: Conceptualization; Supervision; Funding acquisition; Writing—original draft; Writing—review and editing.

Source data underlying figure panels in this paper may have individual authorship assigned. Where available, figure panel/source data authorship is listed in the following database record: biostudies:S-SCDT-10_1038-S44319-024-00300-9.

## Funding

## Disclosure and competing interests statement

The authors declare no competing interests.

# Expanded View Figures

**Figure EV1. Related to Fig. 1. The effect of USP9X-KD on integrin levels.** ▶

(A–H) WB and densitometric quantification of Itga5 and Itgb1 protein levels (**A, C, E, G**) and flow cytometry analysis of Itga5 and Itgb1 surface levels (**B, D, F, H**) in mouse fibroblasts (**A, B**), Hela (**C, D**), RPE-1 (**E, F**), and MDA-MB-231 cells (**G, H**) treated with control non-targeting siRNA (CTL) or siRNAs targeting USP9X (USP9X-KD). Cells were cultured overnight in DMEM with 10% serum or serum-replacement medium. Gapdh served as a loading control. Statistical analysis was carried out by paired *t*-test. In (**A**), statistical significance was tested between the CTL and USP9X-KD groups with or without serum (for Itga5, $P = 0.2973$ and 0.3151, respectively; for Itgb1, $P = 0.1435$ and 0.9821, respectively). In (**B**), statistical significance was tested between the CTL and USP9X-KD groups with or without serum (for Itga5, $P = 0.2412$ and 0.0346, respectively; for Itgb1, $P = 0.6668$ and 0.4548, respectively). In (**C**), statistical significance was tested between the CTL and USP9X-KD groups with or without serum (for Itga5, $P = 0.0573$ and 0.0321, respectively; for Itgb1, $P = 0.8148$ and 0.1067, respectively). In (**D**), statistical significance was tested between the CTL and USP9X-KD groups with or without serum (for Itga5, $P = 0.1340$ and 0.1064, respectively; for Itgb1, $P = 0.0409$ and 0.0622, respectively). In (**E**), statistical significance was tested between the CTL and USP9X-KD groups with or without serum (for Itga5, $P = 0.0110$ and 0.0457, respectively; for Itgb1, $P = 0.1110$ and 0.2666, respectively). In (**F**), statistical significance was tested between the CTL and USP9X-KD groups with or without serum (for Itga5, $P = 0.1336$ and 0.1802, respectively; for Itgb1, $P = 0.0086$ and 0.0620, respectively). In (**G**), statistical significance was tested between the CTL and USP9X-KD groups with or without serum (for Itga5, $P = 0.2786$ and 0.0450, respectively; for Itgb1, $P = 0.0271$ and 0.1004, respectively). In (**H**), statistical significance was tested between the CTL and USP9X-KD groups with or without serum (for Itga5, $P = 0.0117$ and 0.0001, respectively; for Itgb1, $P = 0.0986$ and 0.0954, respectively). *$P < 0.05$; **$P < 0.01$; ***$P < 0.001$; n.s. not significant. Data were shown as Mean ± SD, $n = 3$ independent experiments.

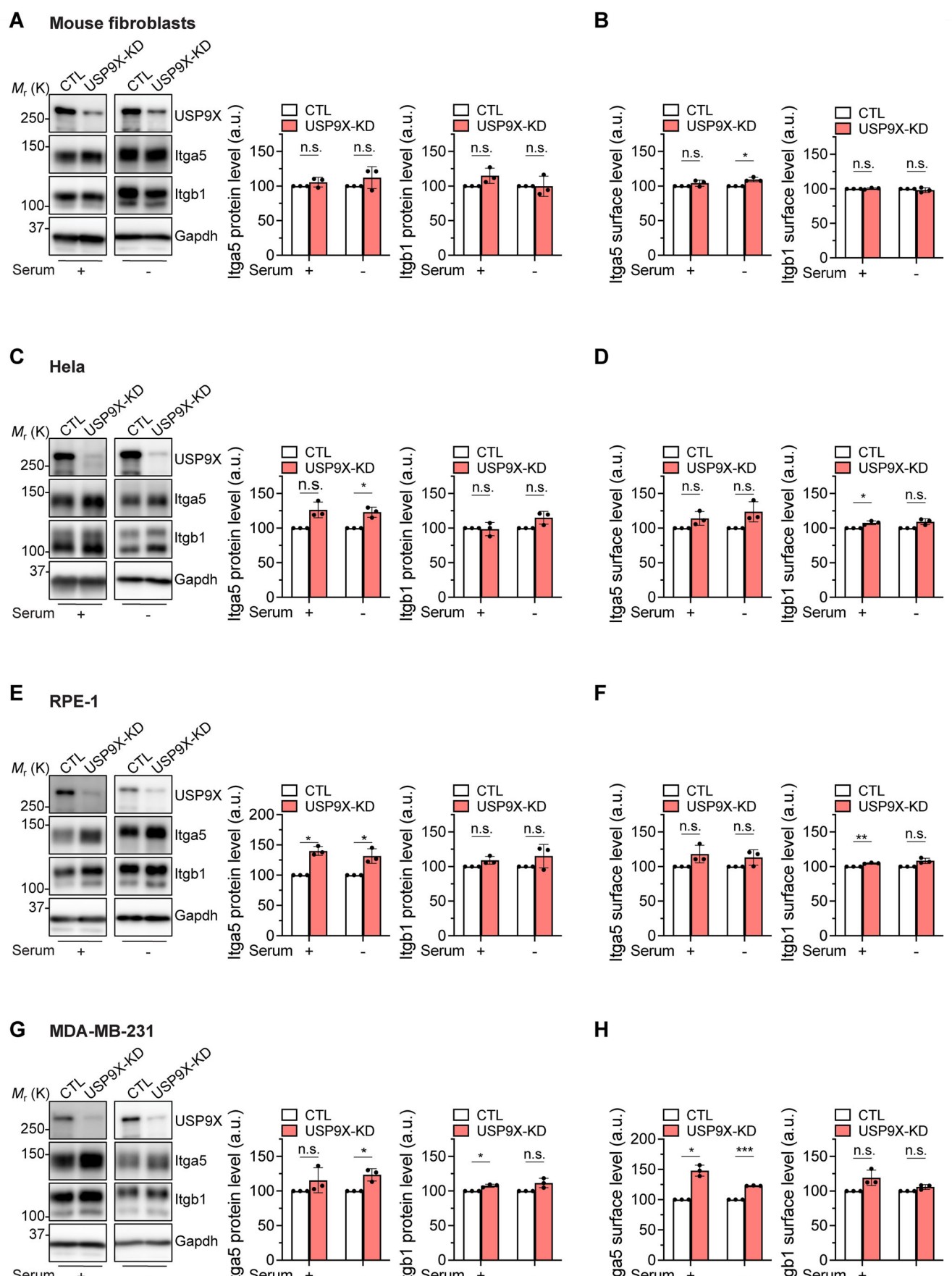

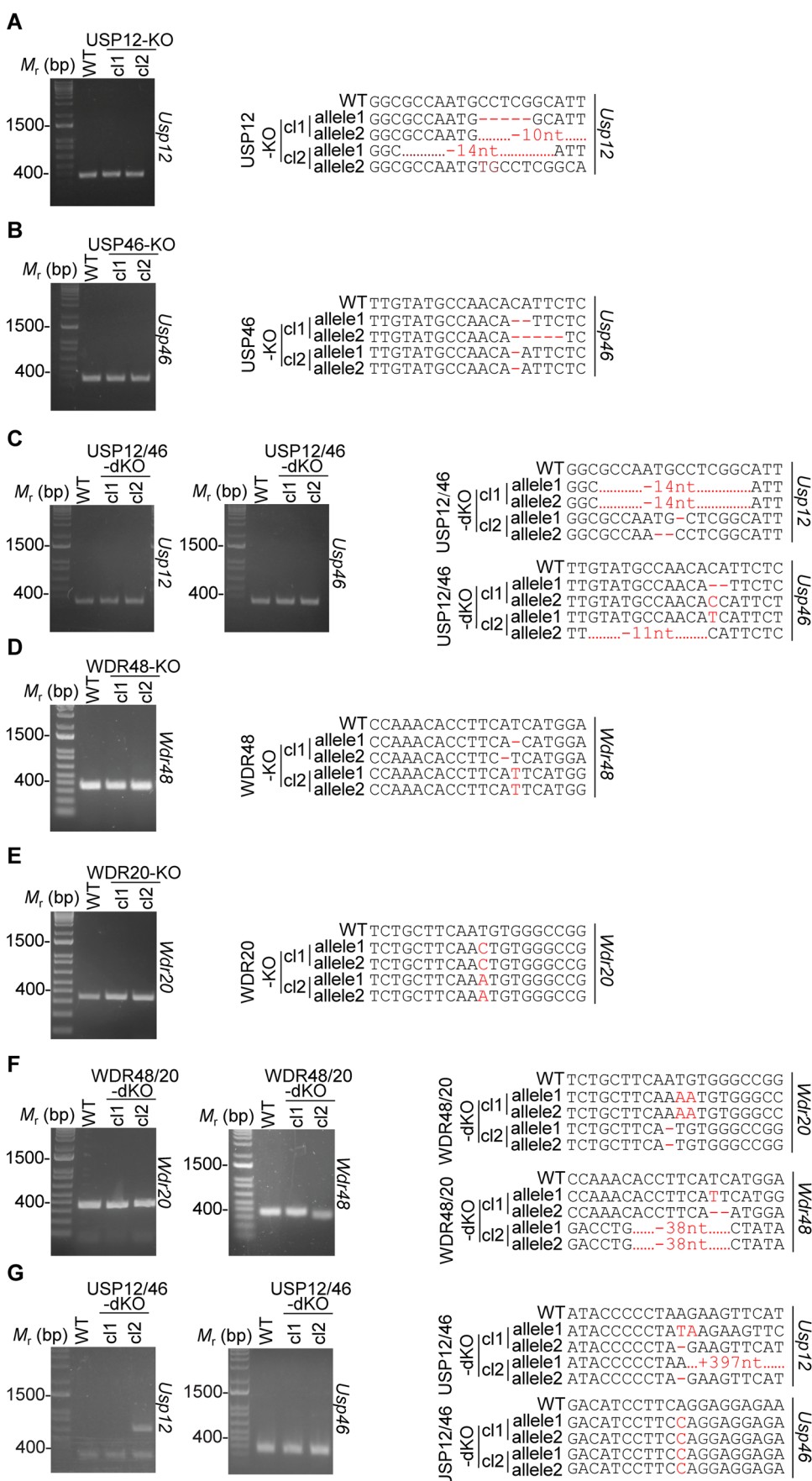

◄ **Figure EV2.   Related to Fig. 1. Validation of KO clones.**

(**A–G**) Agarose gel electrophoresis images show PCR amplification products from the genomic region containing the indicated Cas9 targeting sites of the indicated genes in the parental WT and two independent mouse fibroblast clones (**A–F**) and MDA-MB-231 cell clones (**G**). PCR products were sequenced, analyzed with the Synthego Inference of CRISPR Edits (ICE) analysis tool (Hsiau et al, 2018), and the corresponding alignments are shown on the right panel.

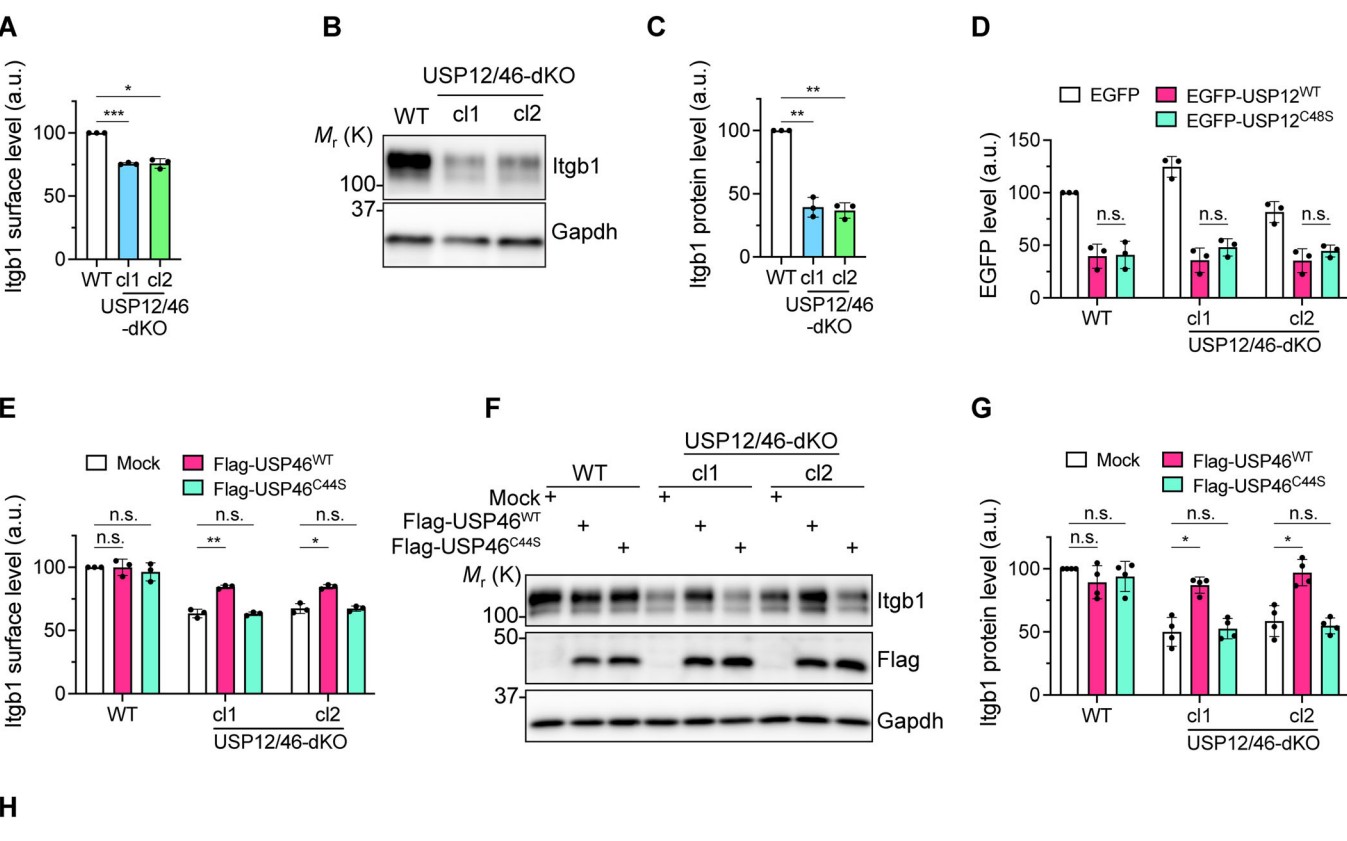

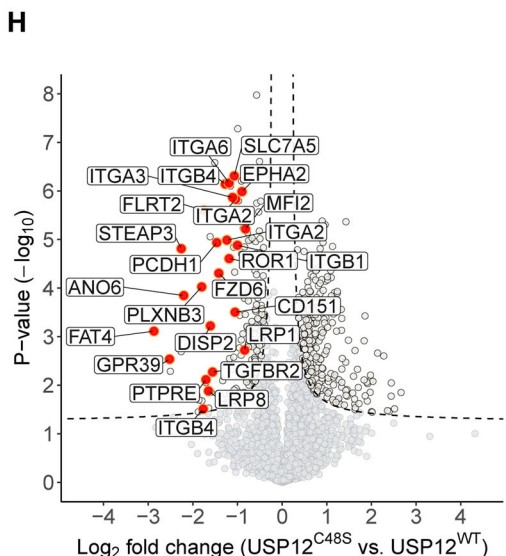

**Figure EV3. Related to Fig. 1. The USP12/46-WDRs complex maintains Itgb1 protein levels.**

(A–C) Itgb1 surface levels determined by flow cytometry (A) and Itgb1 protein levels in cell lysates determined by WB (B) with densitometric quantification (C) in WT and USP12/46-dKO MDA-MB-231 cells. Gapdh served as a loading control. Statistical analysis was carried out by RM one-way ANOVA with Dunnett's multiple comparison test. In (A), statistical significance was tested by comparing WT with USP12/46-dKO cl1 or cl2 MDA-MB-231 cells ($P = 0.0006$ and $0.0126$, respectively). In (C), statistical significance was tested by comparing WT with USP12/46-dKO cl1 or cl2 MDA-MB-231 cells ($P = 0.0084$ and $0.0048$, respectively). *$P < 0.05$; **$P < 0.01$; ***$P < 0.001$. Data were shown as Mean ± SD, $n = 3$ independent experiments. (D) EGFP fluorescence intensities in WT and USP12/46-dKO fibroblasts stably expressing EGFP, EGFP-USP12$^{WT}$, or EGFP-USP12$^{C48S}$ determined by flow cytometry. Statistical analysis was carried out by ordinary two-way ANOVA with Šidák's multiple comparison test comparing the EGFP-USP12$^{WT}$ group and EGFP-USP12$^{C48S}$ group in WT fibroblasts ($P = 0.9987$); in USP12/46-dKO cl1 fibroblasts ($P = 0.4423$) and in USP12/46-dKO cl2 fibroblasts ($P = 0.6692$). n.s. not significance. Data were shown as Mean ± SD, $n = 3$ independent experiments. (E–G) Itgb1 surface levels determined by flow cytometry (E) and Itgb1 protein levels in cell lysates determined by WB (F) with densitometric quantification (G) in WT and USP12/46-dKO fibroblasts stably expressing FLAG-USP46$^{WT}$ or FLAG-USP46$^{C44S}$. Mock-transduced cells (Mock) served as control. Gapdh served as a loading control. Statistical analysis was carried out by RM two-way ANOVA with Dunnett's multiple comparison test. In (E), statistical significance was tested comparing the Mock group with Flag-USP46$^{WT}$ or Flag-USP46$^{C44S}$ group in WT fibroblasts ($P = 0.9990$ and $0.6491$, respectively); in USP12/46-dKO cl1 fibroblasts ($P = 0.0045$ and $0.9669$, respectively); and in USP12/46-dKO cl2 fibroblasts ($P = 0.0329$ and $0.9996$, respectively). In (G), statistical significance was tested comparing the Mock group with Flag-USP46$^{WT}$ or Flag-USP46$^{C44S}$ group in WT fibroblasts ($P = 0.3116$ and $0.5586$, respectively); in USP12/46-dKO cl1 fibroblasts ($P = 0.0127$ and $0.4378$, respectively); and in USP12/46-dKO cl2 fibroblasts ($P = 0.0168$ and $0.7514$, respectively). *$P < 0.05$; **$P < 0.01$; n.s. not significant. Data were shown as Mean ± SD. (E) $n = 3$; (F, G) $n = 4$ independent experiments. (H) Volcano plot of the cell surface proteome of USP12/46-dKO MDA-MB-231 cells expressing EGFP-USP12$^{C48S}$ versus EGFP-USP12$^{WT}$ identified by label-free MS. $P$ values are determined using two-sided permuted $t$-test with 250 randomizations. The black dashed line indicates the significance cutoff (FDR:0.05, S0:0.1) estimated by the Perseus software. $n = 4$ biological replicates. Arbitrarily selected cell surface receptors are highlighted in red.

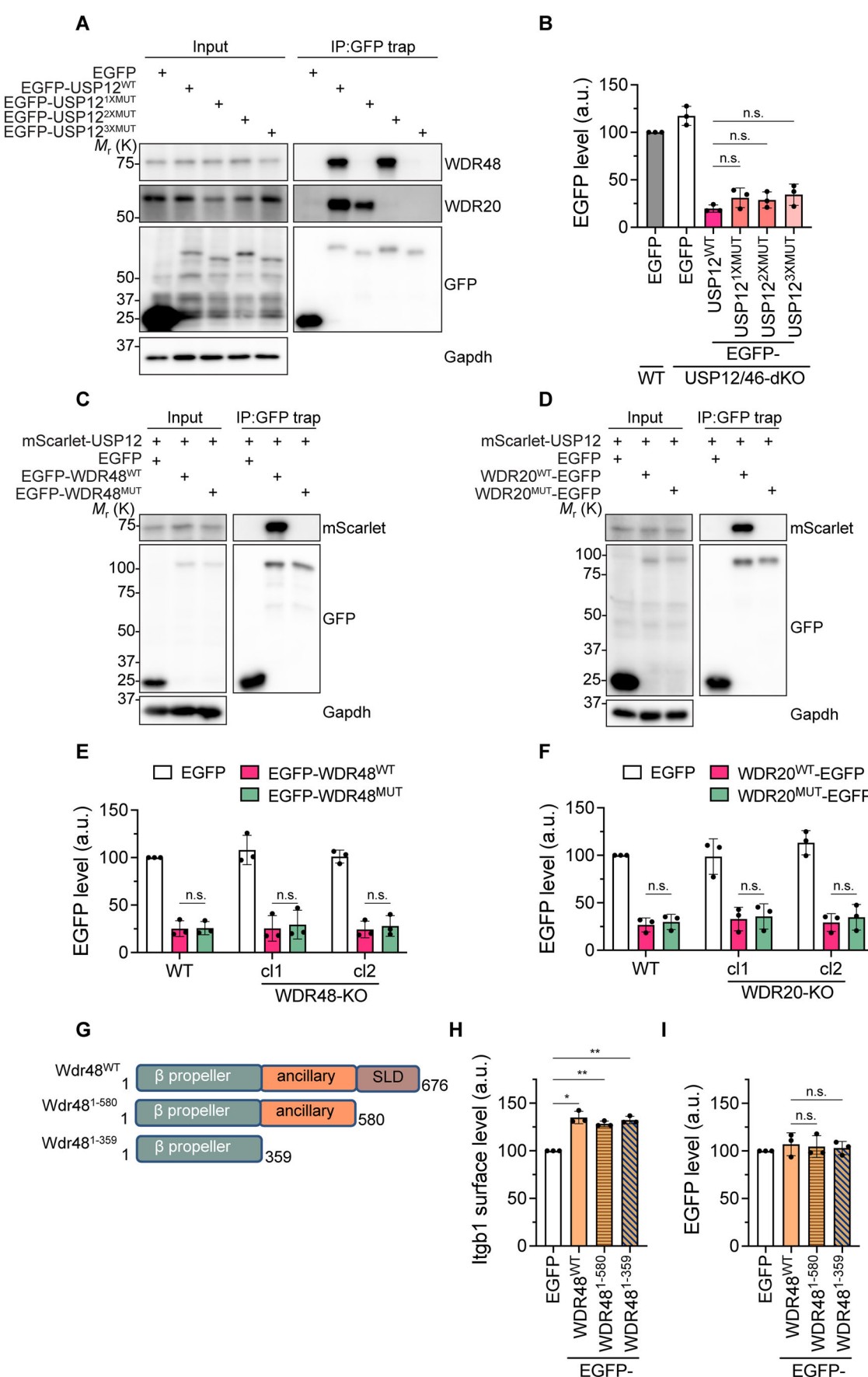

**Figure EV4.** Related to Fig. 2. Characterization of binding-deficient USP12, WDR48, and WDR20 mutants.

(A) GFP immunoprecipitation (GFP IP) from USP12/46-dKO fibroblasts transiently expressing EGFP, EGFP-USP12$^{WT}$, EGFP-USP12$^{1XMUT}$, EGFP-USP12$^{2XMUT}$, or EGFP-USP12$^{3XMUT}$ analyzed by WB for indicated proteins. Gapdh served as a loading control. Representative images from three independent experiments are shown. (B) EGFP fluorescence in WT and USP12/46-dKO fibroblasts transiently expressing EGFP, EGFP-USP12$^{WT}$, EGFP-USP12$^{1XMUT}$, EGFP-USP12$^{2XMUT}$, or EGFP-USP12$^{3XMUT}$ determined by flow cytometry. Statistical analysis was carried out by ordinary one-way ANOVA with Dunnett's multiple comparison test comparing the EGFP-USP12$^{WT}$ group with EGFP-USP12$^{1XMUT}$, EGFP-USP12$^{2XMUT}$, or EGFP-USP12$^{3XMUT}$ group in USP12/46-dKO cl1 fibroblasts ($P = 0.3384$, 0.4963, and 0.1783, respectively). n.s. not significant. Data were shown as Mean ± SD, $n = 3$ independent experiments. (C) GFP IP from USP12/46-dKO fibroblasts stably expressing mScarlet-USP12 and transiently expressing EGFP, EGFP-WDR48$^{WT}$, or EGFP-WDR48$^{MUT}$ analyzed by WB for indicated proteins. Gapdh served as a loading control. Representative images from three independent experiments are shown. (D) GFP IP from USP12/46-dKO fibroblasts stably expressing mScarlet-USP12 and transiently expressing EGFP, WDR20$^{WT}$-EGFP, or WDR20$^{MUT}$-EGFP analyzed by WB for indicated proteins. Gapdh served as a loading control. Representative images from three independent experiments are shown. (E) EGFP fluorescence in WT and WDR48-KO fibroblasts transiently expressing EGFP, EGFP-WDR48$^{WT}$, or EGFP-WDR48$^{MUT}$ determined by flow cytometry. Statistical analysis was carried out by ordinary two-way ANOVA with Šidák's multiple comparison test comparing the EGFP-WDR48$^{WT}$ group with EGFP-WDR48$^{MUT}$ group in WT fibroblasts ($P = 0.9999$); in WDR48-KO cl1 fibroblasts ($P = 0.9630$); and in WDR48-KO cl2 fibroblasts ($P = 0.9701$). n.s. not significant. Data were shown as Mean ± SD, $n = 3$ independent experiments. (F) EGFP fluorescence in WT and WDR20-KO fibroblasts transiently expressing EGFP, WDR20$^{WT}$-EGFP, or WDR20$^{MUT}$-EGFP determined by flow cytometry. Statistical analysis was carried out by ordinary two-way ANOVA with Šidák's multiple comparison test comparing the WDR20$^{WT}$-EGFP group with the WDR20$^{MUT}$-EGFP group in WT fibroblasts ($P = 0.9814$); in WDR20-KO cl1 fibroblasts ($P = 0.9870$); and in WDR20-KO cl2 fibroblasts ($P = 0.9070$). n.s. not significant. Data were shown as Mean ± SD, $n = 3$ independent experiments. (G) Domain organization of the WT WDR48 and WDR48 domain-deletion mutants. (H, I) Itgb1 surface levels (H) and EGFP fluorescence (I) in WDR48-KO fibroblasts stably expressing EGFP, EGFP-WDR48$^{WT}$, EGFP-WDR48$^{1-580}$, or EGFP-WDR48$^{1-359}$ determined by flow cytometry. Statistical analysis was carried out by RM one-way ANOVA with Dunnett's multiple comparison test. In (H), statistical significance was tested comparing the EGFP group with EGFP-WDR48$^{WT}$, EGFP-WDR48$^{1-580}$, or EGFP-WDR48$^{1-359}$ group ($P = 0.0216$, 0.0065, and 0.0093, respectively). In (I), statistical significance was tested by comparing the EGFP-WDR48$^{WT}$ group with EGFP-WDR48$^{1-580}$ or EGFP-WDR48$^{1-359}$ group ($P = 0.8951$ and 0.7411, respectively). $^*P < 0.05$; $^{**}P < 0.01$; n.s. not significant. Data were shown as Mean ± SD, $n = 3$ independent experiments.

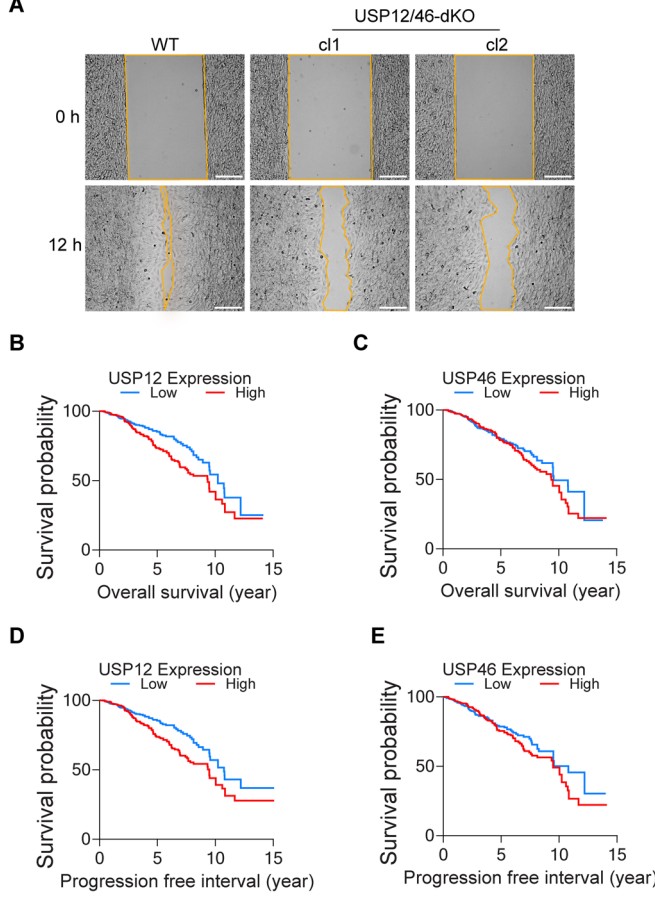

**Figure EV5.** Related to Fig. 8. USP12 and USP46 are not favorable for prognosis in cancer patients.

(**A**) Representative images of the in vitro wound healing assay showing WT and USP12/46-dKO fibroblasts migrating on FN-coated 2D surfaces at 0 and 12 h. Lines mark the leading edge of cell migration towards the wound. Scale bar, 200 μm. (**B–E**) Kaplan–Meier plot of the overall survival (**B, D**) and progression-free interval (**C, E**) of breast cancer patients with high (red line) or low (blue line) gene expression *USP12* (**B, C**) or *USP46* (**D, E**) levels. The GDC TCGA dataset obtained from the UCSC Xena project (Goldman et al, 2020) was used. Two-group risk model with a cut-off at the median was applied. *P* values were calculated by log-rank test. The *P* values in (**B–E**) are 0.0059, 0.3898, 0.0058, and 0.3927, respectively.

