## [Peer Review File · EMBO Reports]

The USP12/46 deubiquitinases protect integrins from ESCRT-mediated lysosomal degradation

Kaikai Yu, Guan Wang, Shiny Shengzhen Guo, Florian Bassermann, and Reinhard Fässler

Corresponding author(s): Reinhard Fässler (faessler@biochem.mpg.de), Reinhard Fässler (faessler@biochem.mpg.de)

Review Timeline:

Submission Date:	14th May 24
Editorial Decision:	24th Jun 24
Revision Received:	12th Aug 24
Editorial Decision:	4th Sep 24
Revision Received:	2nd Oct 24
Accepted:	16th Oct 24

Transaction Report:

Dear Dr. Wang

Thank you for the submission of your research manuscript to our journal. We have now received the full set of referee reports that is copied below.

As you will see, all referees acknowledge that the findings are interesting and that the conclusions are overall supported by the data presented but they also raise a number of concerns and have suggestions how to further strengthen the data, which should be addressed in a revision. Please also note that all Methods must be part of the main manuscript text. Along these lines, I noted that the section on "Integrin degradation, internalization and recycling assays" refers to a previous study without giving any more details. Please carefully check whether you indeed followed exactly this protocol and I recommend adding at least a minimal set of experimental details.

Given the constructive comments and support from the referees, we would like to invite you to revise your manuscript with the understanding that the referee concerns (as detailed above and in their reports) must be fully addressed and their suggestions taken on board. Please address all referee concerns in a complete point-by-point response. Acceptance of the manuscript will depend on a positive outcome of a second round of review. It is EMBO Reports policy to allow a single round of revision only and acceptance or rejection of the manuscript will therefore depend on the completeness of your responses included in the next, final version of the manuscript.

We realize that it is difficult to revise to a specific deadline. In the interest of protecting the conceptual advance provided by the work, we recommend a revision within 3 months (September 24). Please discuss the revision progress ahead of this time with the editor if you require more time to complete the revisions.

I am also happy to discuss the revision further via e-mail or a video call, if you wish.

*******IMPORTANT NOTE:**

We perform an initial quality control of all revised manuscripts before re-review. Your manuscript will FAIL this control and the handling will be delayed IN CASE the following APPLIES:

- 1) A data availability section providing access to data deposited in public databases is missing. If you have not deposited any data, please add a sentence to the data availability section that explains that.
- 2) Your manuscript contains statistics and error bars based on $n=2$. Please use scatter blots in these cases. No statistics should be calculated if $n=2$.

When submitting your revised manuscript, please carefully review the instructions that follow below. Failure to include requested items will delay the evaluation of your revision.*****

- 1) a .docx formatted version of the manuscript text (including legends for main figures, EV figures and tables). Please make sure that the changes are highlighted to be clearly visible.
- 2) individual production quality figure files as .eps, .tif, .jpg (one file per figure). Please download our Figure Preparation Guidelines (figure preparation pdf) from our Author Guidelines pages <https://www.embopress.org/page/journal/14693178/authorguide> for more info on how to prepare your figures.
- 3) a .docx formatted letter INCLUDING the reviewers' reports and your detailed point-by-point responses to their comments. As part of the EMBO Press transparent editorial process, the point-by-point response is part of the Review Process File (RPF), which will be published alongside your paper.
- 4) a complete author checklist, which you can download from our author guidelines (<<https://www.embopress.org/page/journal/14693178/authorguide>>). Please insert information in the checklist that is also reflected in the manuscript. The completed author checklist will also be part of the RPF.

5) Please note that all corresponding authors are required to supply an ORCID ID for their name upon submission of a revised manuscript (<<https://orcid.org/>>). Please find instructions on how to link your ORCID ID to your account in our manuscript tracking system in our Author guidelines (<<https://www.embopress.org/page/journal/14693178/authorguide#authorshipguidelines>>)

6) We replaced Supplementary Information with Expanded View (EV) Figures and Tables that are collapsible/expandable online. A maximum of 5 EV Figures can be typeset. EV Figures should be cited as "Figure EV1, Figure EV2" etc... in the text and their respective legends should be included in the main text after the legends of regular figures.

7) Before submitting your revision, primary datasets (and computer code, where appropriate) produced in this study need to be deposited in an appropriate public database (see <<https://www.embopress.org/page/journal/14693178/authorguide#dataavailability>>). Specifically, we would kindly ask you to provide public access to the mass spectrometry dataset.

The accession numbers and database should be listed in a formal "Data Availability " section (placed after Materials & Method) that follows the model below (see also <<https://www.embopress.org/page/journal/14693178/authorguide#dataavailability>>). Please note that the Data Availability Section is restricted to new primary data that are part of this study.

Data availability

Additional information on source data and instruction on how to label the files are available <<https://www.embopress.org/page/journal/14693178/authorguide#sourcedata>>.

10) Figure legends and data quantification:

- the name of the statistical test used to generate error bars and P values,
 - the number (n) of independent experiments (please specify technical or biological replicates) underlying each data point,
 - the nature of the bars and error bars (s.d., s.e.m.)
- If the data are obtained from n {less than or equal to} 5, show the individual data points in addition to the SD or SEM.
- If the data are obtained from n {less than or equal to} 2, use scatter blots showing the individual data points.

11) Our journal encourages inclusion of *data citations in the reference list* to directly cite datasets that were re-used and obtained from public databases. Data citations in the article text are distinct from normal bibliographical citations and should directly link to the database records from which the data can be accessed. In the main text, data citations are formatted as follows: "Data ref: Smith et al, 2001" or "Data ref: NCBI Sequence Read Archive PRJNA342805, 2017". In the Reference list, data citations must be labeled with "[DATASET]". A data reference must provide the database name, accession number/identifiers and a resolvable link to the landing page from which the data can be accessed at the end of the reference. Further instructions are available at <https://www.embopress.org/page/journal/14693178/authorguide#referencesformat>.

12) All Materials and Methods need to be described in the main text. We would encourage you to use 'Structured Methods', our new Methods format. According to this format, the Methods section should include a Reagents and Tools Table (listing key reagents, experimental models, software and relevant equipment and including their sources and relevant identifiers) followed by a Methods and Protocols section in which we encourage the authors to describe their methods using a step-by-step protocol format with bullet points, to facilitate the adoption of the methodologies across labs. More information on how to adhere to this format as well as downloadable templates (.docx) for the Reagents and Tools Table can be found in our author guidelines: <https://www.embopress.org/page/journal/14693178/authorguide#manuscriptpreparation>.

<https://www.embopress.org/doi/full/10.1038/s44320-024-00037-6#sec-4>.

13) As part of the EMBO publication's Transparent Editorial Process, EMBO Reports publishes online a Review Process File to accompany accepted manuscripts. This File will be published in conjunction with your paper and will include the referee reports, your point-by-point response and all pertinent correspondence relating to the manuscript.

Yours sincerely,

Referee #1:

In their manuscript, Yu and colleagues, by combining genetic screening in haploid Cas9-expressing HAP1 cells and a BioID-based proximity labeling of integrin $\beta 1$ (Itgb1) identify USP46, USP12, WDR48 and WDR2 as the components of the major deubiquitinating complex (DUB) of Itgb1 under steady-state culture conditions. Through an extremely robust series of knockdown (KO) and rescue experiments with wild type (WT) or mutant constructs in at least two different clones of murine fibroblasts and MDA-MB-231 human breast carcinoma cells, the Authors clearly demonstrate how the DUB USP12/46-WDR48-WDR20 complex is critical in enabling the post-endocytic recycling and rescue from polyubiquitin K63-dependent and ESCRT-dependent lysosomal degradation of Itgb1, and the $\alpha 5\beta 1$ heterodimer in particular. The Authors conclude by demonstrating how expression levels of USP12/46, in addition to controlling adhesion, migration and invasion of MDA-MB-231 cells, are related to the overall survival and progression-free interval of breast cancer patients.

The study is clearly and thoroughly developed, identifying a novel and crucial complex that, by allowing the recycling of endocytosed Itgb1, controls Itgb1 half-life and function with important (also pathological) implications. The overall picture could still be completed by two integrations.

1. In a series of experiments, such as those shown in Figure 4H-j, the Authors give evidence that deubiquitylation is necessary for the recycling of endocytosed Itgb1. Indeed, lysosomal inhibition with Bafilomycin A1 restores total Itgb1 levels, but does not recover its physiological levels on the surface of USP12/46-dKO fibroblasts. On the other hand, ESCRT silencing increases the surface levels of $\alpha 5$ WT $\beta 1$ WT, but little of $\alpha 5$ KR $\beta 1$ KR in Itgb1-KO/USP12/46-dKO fibroblasts. It would complete the overall picture of the manuscript if the Authors could offer support for a mechanistic hypothesis of the reason why deubiquitylation is required for subsequent surface recycling of endocytosed Itgb1. Could the ubiquitination of the cytosolic tail of Itgb1 (e.g., at the K794 level of the NPK794Y motif) impair the PTB-domain-mediated interaction with SNX17, while favoring its association with the ESCRT complex? Would this be experimentally testable, e.g., by using the approach described in Böttcher et al., Nat Cell Biol, 2012?

2. It would be helpful if the Authors could comment in the Discussion about whether or not USP12 and USP46 might represent potential targets of a hypothetical pharmacological approach aimed at their inhibition, for example in the context of breast cancer.

Referee #2:

Review of Yu et al (USP46/USP12 and ITGB1 manuscript)

I very much enjoyed reading this very interesting, well-crafted manuscript, which uses a DUB focussed CRISPR based screen in conjunction with proximity biotinylation proteomics to identify the two paralogous DUBs, USP12 and USP46 as direct regulators of lysosomal trafficking of integrins.

Both DUBs form functional ternary complexes with two WD40-proteins WDR48 and WDR20, and the authors provide compelling evidence in form of KO and rescue experiments, for a requirement not just of catalytic activity of the DUBs but also of the presence of and association with these two cofactors. In the absence of both The authors go on to implicate the ESCRT machinery (using an ESCRT-0 and ESCRTI double knock down) and K63-linked ubiquitin chains as mediators of ubiquitin dependent downregulation of ITGB1 and ITGA5. A final set of figures relates the effects of USP12 and 46 KO on integrin expression levels and integrin surface levels to functional adhesion read-outs. The biochemical approaches are supplemented by very informative quantitated immunofluorescence imaging approaches. The manuscript is well written and well argued. Overall this manuscript provides important new mechanistic insights into the trafficking routes and ubiquitin dependent turnover of integrins and informs on the biology of two hitherto poorly characterised DUBs.

I have listed a few minor data-specific comments below:

With regards to Figure 1 and the discussion: The authors do not comment on why they think it was possible for them to identify USP46 in their CRISPR screen, if there is clear redundancy between USP46 and USP12. It may also be useful to point to USP12 in Figure 1B? Is it possible that USP12 expression levels are particularly low in the HAP1 cells?

Figure EV2G cl1 allele 1 and 2 usp46 - should the last base be an A rather than a G or should the G be labelled in red?

Figure 3A and C : in these graphs and associated legend it is unclear what the P-values are related to. If as the figure implies the comparison is made with the WT then the key stats are missing. The text states that the KO of SNX17 on top of USP12/46 has a stronger effect than either 12/46 alone or SNX17 alone.

Figure 4: Please clarify on the x-axis of the graphs in this figure whether the time is referring to a CHX chase and on the Y-axis whether the data refer to total or cell surface levels. This will help the reader. In Figure 4H please state clearly in the text that the treatment was for 9 hours.

General comment on the CHX chases - such long chases are difficult to interpret, and with such few datapoints I would suggest refraining from deducing values for half-lives in the text.

Figure 4J - That BAF does not rescue cell surface levels was not surprising to this reader, but what is more surprising and not remarked on is that it does rescue the SNX17 KO associated decrease in ITGB1 cell surface levels. This may be worthwhile mentioning and exploring in the discussion.

Figure 5 - It would be really helpful if ideally all quantifications that have been relegated to EV5 could be shown in Figure 5. I think that EV5 is essential and should be incorporated in the main figures.

Figure 5D - it is currently very unclear from the text and legends, which cells these mass spec data have been derived from. Is this from a comparison of USP12/46 dKO vs WT lines in each case with ESCRT KD? I also could not find a section for this in the methods. Later on the authors generate Knull mutants of ITGB1 and elect to mutate all Ks. Do the authors have evidence that suggests the three identified Ks in Figure 5D are not the only ones that are ubiquitylated? This choice (of generating a complete Knull) may just be worth rationalising in the result section.

Figure 5G - this is very nice and could become its own figure if space is needed for the quantifications for the blots above. It does look like USP12 and ITGB1 colocalise on enlarged endosomes, but the authors have not provided a colocalisation of either with an endosomal marker. An additional set of panels showing the colocalisation of ITGB1 and USP12 with endosomes would be very helpful - an EEA1 antibody would be appropriate here. Aligned with this: have the authors assessed the localisation of endogenous USP12 or USP46 under these conditions?

It is odd to find no parts of the Methods in the main part of the manuscript.

Line 205-207: "cells. A role for Itgb1 mRNA transcript stability in regulating Itgb1 protein levels could be excluded as no difference in Itgb1 mRNA levels were found between USP12/46-dKO and WT fibroblasts (Fig. 4A). - the p-value for cl2 actually suggests an increase - so suggest replacing "no difference" with "no decrease".

Line 2019-202: "the USP12/46-WDRs complex stabilizes the surface as well as total Itgb1 protein levels" - suggest replacing "stabilizes" with "maintains".

Line 291: "ubiquitinable" - typo

Line 331 - 332 - consider rephrasing.

Line 335 - consider replacing "we fund that" with "we hypothesised that"

Referee #3:

The manuscript by Yu et al describes a robust approach combining a Crispr/Cas9 screen with proximity labelling proteomics to identify DUBs that regulate ITGB1 levels in cells at steady state. The authors convincingly show that USP12 and its paralogue USP46 are DUBs that maintain integrin levels at steady state by removing K63-linked ubiquitin modifications. The USP12/46 associated adaptor proteins WDR48 and WDR20 are also required for maintenance of steady-state integrin levels. Functionally, low levels of USP12/46 decrease cell adhesion and motility, and high levels of USP12 correlate with poor outcomes in breast cancer. This is a very convincing and detailed study that identifies new players in regulation of integrins, and is close to the level and quality required for publication. I have three queries that the authors should address however:

1. The authors state that "SNX17 and USP12/46-WDR complex stabilise ITGB1 independently". Could it not be sequential, whereby USP12/46-WDRs remove ubiquitin to allow SNX17 binding and recycling? Perhaps this doesn't need to be addressed experimentally but warrants discussion. For figure 3 perhaps further statistical analysis of differences between USP versus SNX17 knockdown would reveal significant differences.
2. Figure 5 western blots would benefit from quantification, particularly panels panel A, E and F
3. Previous studies have identified USP9X and USP10 (Gillespie 2017 JCS) as DUBs targeting ubiquitin-modified integrins. The former is addressed, although some differences between the two studies can be observed, but the latter is not. This could easily be remedied with a brief addition to the discussion section, highlighting differences between the studies perhaps.

POINT-TO-POINT RESPONSES TO THE EDITOR'S AND THE REVIEWERS' COMMENTS

EMBOR-2024-59594-T “The USP12/46 deubiquitinases protect integrins from ESCRT-mediated lysosomal degradation”

We appreciate the constructive comments and the fair handling of the manuscript. We addressed each comment and revised the manuscript accordingly.

Comments from the editor:

Please also note that all Methods must be part of the main manuscript text. Along these lines, I noted that the section on "Integrin degradation, internalization and recycling assays" refers to a previous study without giving any more details. Please carefully check whether you indeed followed exactly this protocol and I recommend adding at least a minimal set of experimental details.

We moved the Material and Methods into the main manuscript and included the experimental details of "Integrin degradation, internalization and recycling assays" in the revised Material and Methods Section.

Referee #1:

In their manuscript, Yu and colleagues, by combining genetic screening in haploid Cas9-expressing HAP1 cells and a BioID-based proximity labeling of integrin $\beta 1$ (Itgb1) identify USP46, USP12, WDR48 and WDR2 as the components of the major deubiquitinating complex (DUB) of Itgb1 under steady-state culture conditions. Through an extremely robust series of knockdown (KO) and rescue experiments with wild type (WT) or mutant constructs in at least two different clones of murine fibroblasts and MDA-MB-231 human breast carcinoma cells, the Authors clearly demonstrate how the DUB USP12/46-WDR48-WDR20 complex is critical in enabling the post-endocytic recycling and rescue from polyubiquitin K63-dependent and ESCRT-dependent lysosomal degradation of Itgb1, and the $\alpha 5\beta 1$ heterodimer in particular. The Authors conclude by demonstrating how expression levels of USP12/46, in addition to controlling adhesion, migration and invasion of MDA-MB-231 cells, are related to the overall survival and progression-free interval of breast cancer patients.

The study is clearly and thoroughly developed, identifying a novel and crucial complex that, by allowing the recycling of endocytosed Itgb1, controls Itgb1 half-life and function with important (also pathological) implications. The overall picture could still be completed by two integrations.

1. In a series of experiments, such as those shown in Figure 4H-j, the Authors give evidence that deubiquitylation is necessary for the recycling of endocytosed Itgb1. Indeed, lysosomal inhibition with Bafilomycin A1 restores total Itgb1 levels, but does not recover its physiological levels on the surface of USP12/46-dKO fibroblasts. On the other hand, ESCRT silencing increases the surface levels of $\alpha 5\beta 1$ WT, but little of $\alpha 5\beta 1$ KR in Itgb1-KO/USP12/46-dKO fibroblasts. It would complete the overall picture of the manuscript if the Authors could offer support for a mechanistic hypothesis of the reason why deubiquitylation is required for subsequent surface recycling of endocytosed Itgb1. Could the ubiquitination of the cytosolic tail of Itgb1 (e.g., at the

K794 level of the NPK794Y motif) impair the PTB-domain-mediated interaction with SNX17, while favoring its association with the ESCRT complex? Would this be experimentally testable, e.g., by using the approach described in Böttcher et al., Nat Cell Biol, 2012?

This is an interesting and important request. However, we faced the problem that our peptide facility is unable to produce integrin tail peptides that carry ubiquitin at specific lysine residues for SNX17 pull down experiments. We also considered immunoprecipitating wild type and NP(K794R)Y Itgb1 expressed in Itgb1-null cells and then probe the precipitate for SNX17 binding. The problem of this approach is the low affinity of SNX17 for Itgb1, low amount of ubiquitinated Itgb1 in live cells and the difficulty to co-precipitate a complex consisting of a transmembrane protein such as the Itgb1 bound to a low affinity interactor such as SNX17. We were not successful in the past. Therefore, we decided to use Alphafold to test whether the ubiquitinated Itgb1 tail can be modeled with a bound SNX17. The studies were inconclusive because Alphafold revealed binding depending on the 3D location of the attached ubiquitin. We conclude that we are unable to come up with a conclusive answer at this time. We added a sentence in the revised Discussion.

2. It would be helpful if the Authors could comment in the Discussion about whether or not USP12 and USP46 might represent potential targets of a hypothetical pharmacological approach aimed at their inhibition, for example in the context of breast cancer.

This is indeed an important point. We have discussed that the DUBs are potential targets for cancer treatment. This sentence is highlighted in the revised Discussion.

Referee #2:

Review of Yu et al (USP46/USP12 and ITGB1 manuscript)

I very much enjoyed reading this very interesting, well-crafted manuscript, which uses a DUB focussed CRISPR based screen in conjunction with proximity biotinylation proteomics to identify the two paralogous DUBs, USP12 and USP46 as direct regulators of lysosomal trafficking of integrins.

Both DUBs form functional ternary complexes with two WD40-proteins WDR48 and WDR20, and the authors provide compelling evidence in form of KO and rescue experiments, for a requirement not just of catalytic activity of the DUBs but also of the presence of and association with these two cofactors. In the absence of both The authors go on to implicate the ESCRT machinery (using an ESCRT-0 and ESCRTI double knock down) and K63-linked ubiquitin chains as mediators of ubiquitin dependent downregulation of ITGB1 and ITGA5. A final set of figures relates the effects of USP12 and 46 KO on integrin expression levels and integrin surface levels to functional adhesion read-outs. The biochemical approaches are supplemented by very informative quantitated immunofluorescence imaging approaches. The manuscript is well written and well argued. Overall this manuscript provides important new mechanistic insights into the trafficking routes and ubiquitin dependent turnover of integrins and informs on the biology of two hitherto poorly characterised DUBs.

I have listed a few minor data-specific comments below:

With regards to Figure 1 and the discussion: The authors do not comment on why they think it was possible for them to identify USP46 in their CRISPR screen, if there is clear redundancy between USP46 and USP12. It may also be useful to point to USP12 in Figure 1B? Is it possible that USP12 expression levels are particularly low in the HAP1 cells?

The USP12 levels are indeed very low in HAP1 cells. Data from the Human Protein Atlas show that the expression level of USP46 is 1.7-fold higher than that of USP12 in HAP1 cells. It is possible that the lower abundance of USP12 do not suffice to compensate for the loss of USP46. It is also possible that depletion of USP12 was incomplete. We added a comment in revised Result Section.

Figure EV2G c11 allele 1 and 2 usp46 - should the last base be an A rather than a G or should the G be labelled in red?

Last base is indeed an A and we have changed it in the new Figure EV2G. Thank you for pointing out this mistake.

Figure 3A and C : in these graphs and associated legend it is unclear what the P-values are related to. If as the figure implies the comparison is made with the WT then the key stats are missing. The text states that the KO of SNX17 on top of USP12/46 has a stronger effect than either 12/46 alone or SNX17 alone.

Indeed, the comparison was made with the WT fibroblasts versus USP12/46-dKOs or SNX17 KO or SNX17/USP12/46 tKO fibroblasts. We compared the effect of SNX17 KO on top of USP12/46 dKO and revised Figure 3A and C and legends accordingly.

Figure 4: Please clarify on the x-axis of the graphs in this figure whether the time is referring to a CHX chase and on the Y-axis whether the data refer to total or cell surface levels. This will help the reader. In Figure 4H please state clearly in the text that the treatment was for 9 hours.

We revised the labels of the Fig. 4 as suggested by the reviewer.

General comment on the CHX chases - such long chases are difficult to interpret, and with such few datapoints I would suggest refraining from deducing values for half-lives in the text.

We rephrased the text in the Results and Discussion Sections by changing “half-life” to “protein stability”.

Figure 4J - That BAF does not rescue cell surface levels was not surprising to this reader, but what is more surprising and not remarked on is that it does rescue the SNX17 KO associated decrease in ITGB1 cell surface levels. This may be worthwhile mentioning and exploring in the discussion.

We revised the manuscript by adding the following sentences in the Results section: “while BafA1 treatment normalized Itgb1 surface levels in SNX17-KO cells (Böttcher et al., 2012).”

Figure 5 - It would be really helpful if ideally all quantifications that have been relegated to EV5 could be shown in Figure 5. I think that EV5 is essential and should be incorporated in the main figures.

We incorporated quantifications in EV5 to Figure 5 and split Figure 5 into new Figure 5 and 6 in the revised manuscript.

Figure 5D - it is currently very unclear from the text and legends, which cells these mass spec data have been derived from. Is this from a comparison of USP12/46 dKO vs WT lines in each case with ESCRT KD? I also could not find a section for this in the methods. Later on the authors generate Knull mutants of ITGB1 and elect to mutate all Ks. Do the authors have evidence that suggests the three identified Ks in Figure 5D are not the only ones that are ubiquitylated? This choice (of generating a complete Knull) may just be worth rationalising in the result section.

The MS data depicted in Figure 5D is from USP12/46-dKO^{ESCRT-KD} fibroblasts. We added this information in the Figure Legends and Methods Section to improve clarity.

In previous studies we found that mutating individual or groups of lysine residues (even up to seven) in the Itgb1-tail does not stabilize surface Itgb1 levels. Only when all 12 lysine residues in the α 5- and β 1-tails are substituted for arginine residues Itgb1 surface levels become stabilized. (Böttcher *et al.*, 2012). We mention this in the revised Results Section.

Figure 5G - this is very nice and could become its own figure if space is needed for the quantifications for the blots above. It does look like USP12 and ITGB1 colocalise on enlarged endosomes, but the authors have not provided a colocalisation of either with an endosomal marker. An additional set of panels showing the colocalisation of ITGB1 and USP12 with endosomes would be very helpful - an EEA1 antibody would be appropriate here. Aligned with this: have the authors assessed the localisation of endogenous USP12 or USP46 under these conditions?

We thank the reviewer for this very positive comment. As suggested, we added additional panels showing the co-localization of Itgb1 and USP12 with EEA1 (see new Figure 6 H-K and the corresponding Figure Legends) and described the findings in the revised Result Section.

We purchased 3 different polyclonal antibodies to immunostain endogenous USP12 and USP46, in WT and USP12/46 dKO fibroblasts with or without ESCRT-KD (Fig. 1 for the reviewers, see below) and in USP12/46 dKO fibroblasts expressing EGFP-USP12 (Fig. 2 for the reviewers, see below). Unfortunately, none of them showed convincing signals that would have allowed us to assess the localization of the endogenous USP12/46.

A, anti-USP12, SAB1300639, Sigma, 1:50 dilution

B, anti-USP12, 12608-1-AP, Proteintech, 1:100 dilution

C, anti-USP46, 13502-1-AP, Proteintech, 1:100 dilution

Fig. 1 for the reviewers. (A-C) Representative Structured Illumination Microscopy (SIM) images of endogenous USP12 or USP46, Itgb1 and Ub in WT and USP12/46 dKO fibroblasts treated with or without ESCRT-KD siRNAs. Boxes indicate regions displayed in the Zoom panel. Arrowheads indicate the enlarged endosome structures. Scale bar, 10 μ m.

Fig. 2 for the reviewers. (A-C) Representative SIM images of endogenous USP12 or USP46 and EGFP-USP12 in USP12/46-dKO fibroblasts stably expressing EGFP-USP12 treated with or without ESCRT-KD siRNAs. Boxes indicate regions displayed in the Zoom panel. Arrowheads indicate the enlarged endosome structures. Scale bar, 10 μ m.

It is odd to find no parts of the Methods in the main part of the manuscript.

We moved the Material and Methods into the manuscript.

Line 205-207: "cells. A role for Itgb1 mRNA transcript stability in regulating Itgb1 protein levels could be excluded as no difference in Itgb1 mRNA levels were found between USP12/46-dKO and WT fibroblasts (Fig. 4A). - the p-value for cl2 actually suggests an increase - so suggest replacing "no difference" with "no decrease".

We have replaced "no difference" with "did not decrease" as suggested by the reviewer.

Line 2019-202: "the USP12/46-WDRs complex stabilizes the surface as well as total Itgb1 protein levels" - suggest replacing "stabilizes" with "maintains".

We replaced "stabilizes" with "maintains" throughout the text.

Line 291: "ubiquitable" – typo

We changed to 'ubiquitinated'.

Line 331 - 332 - consider rephrasing.

We rephrased the sentence.

Line 335 - consider replacing "we fund that" with "we hypothesised that"

We changed "we found that" to "we hypothesized that"

Referee #3:

The manuscript by Yu et al describes a robust approach combining a Crispr/Cas9 screen with proximity labelling proteomics to identify DUBs that regulate ITGB1 levels in cells at steady state. The authors convincingly show that USP12 and its paralogue USP46 are DUBs that maintain integrin levels at steady state by removing K63-linked ubiquitin modifications. The USP12/46 associated adaptor proteins WDR48 and WDR20 are also required for maintenance of steady-state integrin levels. Functionally, low levels of USP12/46 decrease cell adhesion and motility, and high levels of USP12 correlate with poor outcomes in breast cancer. This is a very convincing and detailed study that identifies new players in regulation of integrins, and is close to the level and quality required for publication. I have three queries that the authors should address however:

1. The authors state that "SNX17 and USP12/46-WDR complex stabilise ITGB1 independently". Could it not be sequential, whereby USP12/46-WDRs remove ubiquitin to allow SNX17 binding and recycling? Perhaps this doesn't need to be addressed experimentally but warrants discussion. For figure 3 perhaps further statistical analysis of differences between USP versus SNX17 knockdown would reveal significant differences.

The two points are also raised by referee #1 and #2. The response to point 1 is as follows:

This is an interesting and important request. However, we faced the problem that our peptide facility is unable to produce integrin tail peptides that carry ubiquitin at specific lysine residues for SNX17 pull down experiments. We also considered immunoprecipitating wild type and NP(K794R)Y Itgb1 expressed in Itgb1-null cells and then probe the precipitate for SNX17 binding. The problem of this approach is the low affinity of SNX17 for Itgb1, low amount of ubiquitinated Itgb1 in live cells and the difficulty to co-precipitate a complex consisting of a transmembrane protein such as the Itgb1 bound to a low affinity interactor such as SNX17. We were not successful in the past. Therefore, we decided to use AlphaFold to test whether the ubiquitinated Itgb1 tail can be modeled with a bound SNX17. The studies were inconclusive because AlphaFold revealed binding depending on the 3D location of the attached ubiquitin. We conclude that we are unable to come up with a conclusive answer at this time. We added a sentence in the revised Discussion.

The response to point 2 is as follows:

Indeed, the comparison was made with the WT fibroblasts versus USP12/46-dKOs or SNX17 KO or SNX17/USP12/46 tKO fibroblasts. We compared the effect of SNX17 KO on top of USP12/46 dKO and revised figure 3A and C and legend accordingly.

2. Figure 5 western blots would benefit from quantification, particularly panels panel A, E and F
As shown in the revised Figure 5B, F and H, we have included quantifications for previous panel A, E and F as suggested.

3. Previous studies have identified USP9X and USP10 (Gillespie 2017 JCS) as DUBs targeting ubiquitin-modified integrins. The former is addressed, although some differences between the two studies can be observed, but the latter is not. This could easily be remedied with a brief addition to the discussion section, highlighting differences between the studies perhaps.

We are grateful for the suggestion of the reviewer. We acknowledge the significance of the study which identified USP10 as a DUB targeting integrins (Gillespie *et al*, 2017). We added a brief addition to the revised Discussion Section.

References:

- Böttcher RT, Stremmel C, Meves A, Meyer H, Widmaier M, Tseng HY, Fässler R (2012) Sorting nexin 17 prevents lysosomal degradation of beta1 integrins by binding to the beta1-integrin tail. *Nat Cell Biol* 14: 584-592
- Gillespie SR, Tedesco LJ, Wang L, Bernstein AM (2017) The deubiquitylase USP10 regulates integrin beta1 and beta5 and fibrotic wound healing. *J Cell Sci* 130: 3481-3495

Dear Dr. Wang

Thank you for the submission of your revised manuscript to EMBO Reports. All three referees had supported publication of your manuscript after minor revisions. You have addressed these concerns in your point-by-point response and by introducing the requested modifications in the manuscript and figures. I am therefore writing with an 'accept in principle' decision, which means that I will be happy to accept your manuscript for publication once you have addressed the editorial requests listed below:

- Regarding the Author Contributions, we now use CRediT to specify the contributions of each author in the journal submission system. Therefore, please remove the Author Contributions from the manuscript file and make sure that the author contributions in our online manuscript tracking system are correct and up-to-date. The information you specified in the system will be automatically retrieved and typeset into the article. You can enter additional information in the free text box provided, if you wish.

- Please update the 'Conflict of interest' paragraph to our new 'Disclosure and competing interests statement'. For more information see <https://www.embopress.org/page/journal/14693178/authorguide#conflictsofinterest>

- Please rename the Supplementary table EV1 to "Table EV1" and Supplementary Table EV2 to "Dataset EV2", both in their legends and their callouts in the text. Dataset EV2 should be uploaded as file type "Dataset".

- Since July 1st all our articles should use our Structured Methods format. To comply with this format, please download and fill our Reagents and Tools Table template (.docx), which you can find in our author guidelines: <https://www.embopress.org/page/journal/14693178/authorguide#structuredmethods>.

- Please note that all data you base conclusions on need to be included in the manuscript. Statements based on "Data not shown" or "unpublished" do not comply with our editorial policies. Therefore, please either remove the statements (page 13) or show the relevant data.

- Please add callouts to the individual panels of Fig EV1 and Fig EV2. Currently, the callouts only refer to "Fig. EV1" without specifying the panels.

- Table 1 should be moved after the main figure legends (or removed if you insert this information in the Reagents and Tools table).

- Please add the heading "Expanded View Figure Legends"

- Data availability section: Please provide the specific URLs that resolve directly to the PXD054760 and PRJNA1146728 datasets, respectively. The reviewer access data need to be removed.

- Author Checklist: you could fill "Cell materials - Report if the cell lines were recently authenticated.... and tested for mycoplasma contamination" since you state in the Methods that you did test regularly for mycoplasma. For "Plants and Microbes/Microbes" please chose YES instead of N/A.

- Our production/data editors have asked you to clarify several points in the figure legends (see below). Please incorporate these changes in the manuscript and return the revised file with tracked changes with your final manuscript submission.

A) Statistical test information. Only p-values that are actually shown in the figure panel(s) should (and must) be defined in the legends, all others should be removed from (or added to) the legend. Moreover, we ask for the specification of exact p-values:
1. Please note that the exact p values are not provided in the legends of figures 2e; 3a; 4d, j, l; 5f, h; 7c-e; EV 1h; EV 3a.

B) Data presentation:

1. Please note that "+1, +2, +3, +4" are not defined in the legend of figure 6c. This needs to be rectified.

2. Please note that the white/pink arrows/ arrowheads are not defined in the legend of figure 4k; 6d, h; 7b. This needs to be rectified.

- We perform a routine screen on all figures and .xls files. Here, we noticed that in two .xls files the exact same numbers (up to the 5th decimal) appear for 4 quantification points. Please double-check whether this is fine or whether there might have occurred a copy and paste error. See color-coded .xls files attached.

- Please describe your findings in the abstract in present tense.

- Finally, EMBO Reports papers are accompanied online by

A) a short (1-2 sentences) summary of the findings and their significance,

B) 2-3 bullet points highlighting key results and

C) a schematic summary figure that provides a sketch of the major findings (not a data image).

Please provide the summary figure as a separate file in PNG or JPG format at a size of 550x300-600 pixels (width x height).

Please note that the size is rather small and that text needs to be readable at the final size. Please send us this information along with the revised manuscript.

- On a different note, I would like to alert you that EMBO Press offers a new format for a video-synopsis of work published with us, which essentially is a short, author-generated film explaining the core findings in hand drawings, and, as we believe, can be very useful to increase visibility of the work. This has proven to offer a nice opportunity for exposure i.p. for the first author(s) of the study. Please see the following link for representative examples and their integration into the article web page:

<https://www.embopress.org/doi/full/10.15252/emj.2019103932>

With kind regards,

Martina Rembold, PhD

Senior Editor

EMBO reports

All editorial and formatting issues were resolved by the authors.

Prof. Reinhard Fässler
Max-Planck-Institute of Biochemistry
Molecular Medicine
Am Klopferspitz 18
Martinsried 82152
Germany

Dear Reinhard,

I am very pleased to accept your manuscript for publication in the next available issue of EMBO reports. Thank you for your contribution to our journal.

Kind regards,

Martina
